**Data Availability Statement:** All relevant data are within the manuscript and its Supporting information files.

# Effectiveness of electrophysical modalities in the sensorimotor rehabilitation of radial, ulnar, and median neuropathies: A meta-analysis

**Ena Bula-Oyola**[1,2]*, **Juan-Manuel Belda-Lois**[3,4], **Rosa Porcar-Seder**[3], **Álvaro Page**[3,5]

**1** Universitat Politècnica de València, Valencia, Spain, **2** Departamento de Diseño, Universidad del Norte, Barranquilla, Colombia, **3** Instituto Universitario de Ingeniería Mecánica y Biomecánica, Universitat Politècnica de València, Valencia, Spain, **4** Grupo de Tecnología Sanitaria del IBV, CIBER de Bioingeniería, Biomateriales y Nanomedicina, Valencia, Spain, **5** Departamento de Física Aplicada, Universitat Politècnica de València, Valencia, Spain

* oyolae@uninorte.edu.co

## Abstract

### Introduction

People with ulnar, radial or median nerve injuries can present significant impairment of their sensory and motor functions. The prescribed treatment for these conditions often includes electrophysical therapies, whose effectiveness in improving symptoms and function is a source of debate. Therefore, this systematic review aims to provide an integrative overview of the efficacy of these modalities in sensorimotor rehabilitation compared to placebo, manual therapy, or between them.

### Methods

We conducted a systematic review according to PRISMA guidelines. We perform a literature review in the following databases: Biomed Central, Ebscohost, Lilacs, Ovid, PEDro, Sage, Scopus, Science Direct, Semantic Scholar, Taylor & Francis, and Web of Science, for the period 1980–2020. We include studies that discussed the sensorimotor rehabilitation of people with non-degenerative ulnar, radial, or median nerve injury. We assessed the quality of the included studies using the Risk of Bias Tool described in the Cochrane Handbook of Systematic Reviews of Interventions and the risk of bias across studies with the GRADE approach described in the GRADE Handbook.

### Results

Thirty-eight studies were included in the systematic review and 34 in the meta-analysis. The overall quality of evidence was rated as low or very low according to GRADE criteria.

Low-level laser therapy and ultrasound showed favourable results in improving symptom severity and functional status compared to manual therapy. In addition, the low level laser showed improvements in pinch strength compared to placebo and pain (VAS) compared to

**Funding:** EBO: scholarship from the Colombian Ministry of Science: "Formación de capital humano de alto nivel para las regiones- Atlántico 2018". URL: https://minciencias.gov.co/convocatorias/oportunidades-formacion/formacion-capital-humano-alto-nivel-para-las-regiones The funders had no role in study design, data collection and analysis, decision to publish, or preparation of the manuscript.

**Competing interests:** The authors have declared that no competing interests exist.

manual therapy. Splints showed superior results to electrophysical modalities. The clinical significance of the results was assessed by effect size estimation and comparison with the minimum clinically important difference (MCID).

## Conclusions

We found favourable results in pain relief, improvement of symptoms, functional status, and neurophysiological parameters for some electrophysical modalities, mainly when applied with a splint. Our results coincide with those obtained in some meta-analyses. However, none of these can be considered clinically significant.

## Trial registration

PROSPERO registration number CRD42020168792; https://www.crd.york.ac.uk/PROSPERO/display_record.php?RecordID=168792.

## Introduction

Peripheral neuropathies are common pathologies. The incidence is up to 300,000 cases per year in Europe and approximately 200,000 in the United States [1]. Peripheral nerves can be damaged by autoimmune or metabolic disorders, tumours, thermal, chemical, or mechanical trauma. The leading causes are penetration, crushing, or pulling, and ischemia [2]. Most of them involve the upper limbs [3], with a higher rate of damage to the ulnar nerve, followed by the median and radial nerves [4,5]. Signs and symptoms may include partial or total motor dysfunction of the forearm and hand, loss of muscle tone and strength, hypoesthesia or hyperesthesia, pain, allodynia, or paraesthesia [6].

Rehabilitation of peripheral neuropathies has surgical and conservative alternatives. Generally, conservative treatment is considered the first option for mild to moderate injuries, while surgical treatment is standard for severe injuries or lesions that do not respond adequately to conservative management [7].

Current literature has focused on the efficacy of surgical and pharmacological treatments [8–21]. Regarding conservative treatments, most research evaluates the effects of electrophysical modalities (EM) in carpal tunnel syndrome (CTS) [22–32]. There is a gap in the study of entrapment injuries in other nerves and pathologies that represent a higher degree of disability, such as paralysis. Despite the available studies, there is no consensus about EM's effects on improving symptoms and function. Therefore, this systematic review aims to provide a comprehensive overview of these therapies' performance in sensorimotor rehabilitation of ulnar, radial, and median neuropathies compared to placebo, physical therapy, or between them.

## Methods

We conducted a systematic review according to PRISMA guidelines (see S1 and S2 Tables). We registered our review in the PROSPERO database for systematic reviews (PROSPERO registration number CRD42020168792) and included the protocol in S1 File.

### Eligibility criteria

The eligibility criteria involved studies published in English over the last forty years evaluating the effectiveness of electrophysical modalities to treat radial, ulnar, or median neuropathies.

The exclusion criteria left out studies that included surgical or pharmacological treatment, animal testing, electrophysical modalities to treat other neuropathies, degenerative neuropathies, or other diseases of diverse origin with neuropathic or musculoskeletal involvement.

## Outcomes measures

The primary outcomes of interest were scores on the pain scale, symptom severity, and functional status. As well as the electrophysiological parameters of the nerves: motor latency, the amplitude of motor action potential, motor conduction velocity, sensory latency, the amplitude of sensory action potential, and sensory conduction velocity. The secondary outcomes were grip and pinch strength.

## Search strategy

We carried out the literature review between April and July 2019 and October 2020, using medical topic headings (MeSH) and free-text terms for neuropathies and rehabilitation to identify studies from the following databases: Biomed Central, Ebscohost, Lilacs, Ovid, PEDro, Sage, Scopus, Science Direct, Semantic Scholar, Taylor & Francis, and Web of Science. We also hand-searched the references from the studies included in the review to find other possible relevant studies. We provide an example of the search terms in the S2 File.

## Data collection and analysis

**Selection of studies and data extraction.** Two independent reviewers (JBL, EBO) examined all articles eligible for inclusion. We classified the data in an Excel matrix according to the type of study; nerve examined, type of injury, severity, characteristics of the participants (number, age, and sex), follow-up periods, intervention, and comparator.

**Assessment of risk of bias.** Two independent reviewers (RPS, AP) assessed the bias of included studies with the Cochrane Risk of Bias tool in five domains: sequence generation, allocation concealment, blinding, incomplete data, and selective information [33]. We resolved disagreements through discussion; in cases where we did not reach a consensus, we consulted a third reviewer (JBL).

**Data synthesis.** We used R Studio software to perform the meta-analysis. We pooled study results according to interventions, outcome measures, and timing of outcome measures. We did the data synthesis for each comparison group separately. In cases where it was not possible to pool the data in a meta-analysis, we provide a narrative synthesis of the results.

We assessed heterogeneity among studies using the I-squared ($I^2$) test. We define heterogeneity using the following ranges as a guide: 0% to 40% might not be important heterogeneity, 30% to 60% might represent moderate heterogeneity, 50% to 90% might represent substantial heterogeneity, and 75% to 100% might represent considerable heterogeneity [33].

We estimated the pooled effect using standardised mean differences (SMDs) with 95% confidence intervals (CI). We used the random-effects model to perform meta-analysis when $I^2 > 50\%$ and the fixed-effects model when $I^2 < 50\%$. We assessed the effect size using Cohen's d coefficient [34] according to the following parameters: <0.2 = trivial effect; 0.2–0.5 = small effect; 0.5–0.8 = moderate effect; > 0.8 = large effect. We used a funnel plot to evaluate publication bias when we could pool at least ten studies examining the same treatment comparison [33].

We used the GRADE approach to summarise the overall quality of evidence for each outcome [35]. According to the GRADE considerations, we assess the quality of evidence across studies: risk of bias, inconsistency, indirect evidence, imprecision, and other considerations (including publication bias, large effect, plausible confounding, and dose-response gradient).

We used GRADEpro GDT software (gradepro.org/) for the assessment and generation of summary tables. We provide footnotes to explain decisions to downgrade or upgrade the quality of evidence. The results of the risk of bias across studies are available in S3 Table.

## Results

### Search strategy

The search strategy yielded 136 results. After eliminating the duplicates, we identified 99 articles. In obtaining the full texts, we excluded several trials: thirteen per language, 42 because the approach was another therapeutic modality (e.g., acupuncture, peloid, kinesiotaping, and paraffin), three that reviewed post-surgical treatments, one whose comparator was no treatment, and two because they included steroid or vitamin B6 injection among their groups (Fig 1).

### Study characteristics

We identified thirty-eight studies evaluating the effectiveness of at least one EM to treat peripheral neuropathies. Thirty-four RCTs (n = 1766) assessed the effects in CTS [36–69], two (n = 93) in ulnar neuropathy at the elbow (UNE) [70,71], one comparative study (n = 19) in radial nerve palsy [72], and another (n = 107) in brachial, median, ulnar, and radial nerve palsy [73]. The characteristics and main outcomes of each study are described in S5 Table.

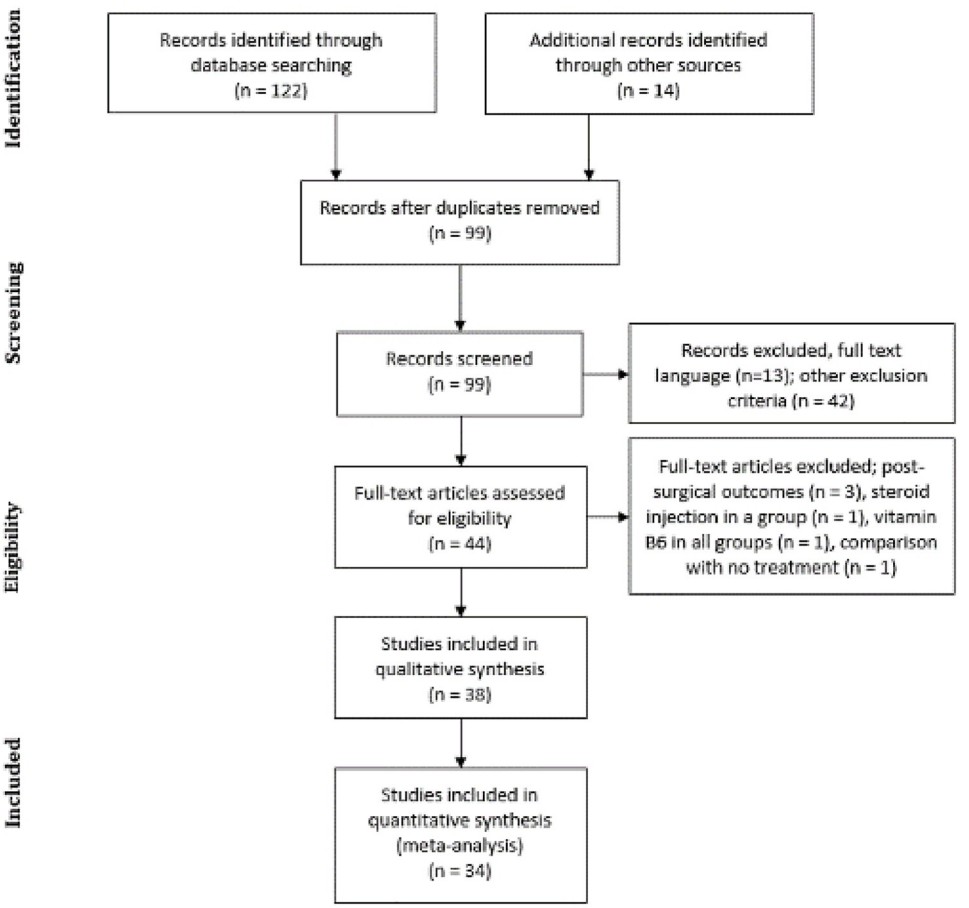

**Fig 1. Flowchart of the study selection process.**

## Assessment of risk of bias

All studies reported that participants were randomly assigned between groups, except one due to diversity between treatments [47]. However, the methods of allocation were not described in some of the studies [36,40,43,44,46,52,59,61,65]. We identified possible performance and detection biases in several studies associated with allocation concealment [37,39,43,44,47,50,51,53,55–57,63,67,68], blinding of participants [37,41–43,46,47,50–52,55–57,62,66–70], blinding of personnel [37,38,41–47,50–53,55–57,60,62,63,66–68,70,71], blinding of outcome assessors [39,42–44,46,49,55,58,60,67] that was unclear or not provided. As well as attrition [36,37,39,40,43,45,48–50,52,54,55,58,60,63,64,70,71] and data reporting biases [48,50,63,65]. In general, all studies had similar baseline characteristics and follow-up times among their groups. The results of the risk of bias assessment of the included studies are available in S4 Table.

## Effects of electrophysical interventions

We obtained thirty-four RCTs evaluating CTS, eighteen comparing EMs versus placebo (i.e., LLLT alone [42,43,48,53,62,63], LLLT plus splint [49,59], ESWT plus splint [38,58], continuous US alone [39], continuous and pulsed US plus splint [45], continuous and pulsed SWD plus splint [37], SMF [40,69], PMF [61] or alternate use of both MF [44]). Four studies assessed EMs against manual therapy (MT) (i.e., LLLT [46,47], LLLT plus US [64], US plus splint, and US with MT plus splint [57]).

Six studies compared different EMs (i.e., LLLT vs. TENS [68], LLLT vs. PMF [67], LLLT vs. pulsed US [41,65], LLLT plus splint vs. continuous plus splint [52], ESWT vs. pulsed US vs. Cryo US [66], TENS vs. IFC [51]), and six studies evaluated EM versus splinting (i.e., LLLT plus splint [50,54], TENS and IFC [51], PRF plus splint [52], ESWT plus splint [55], PPNL plus splint [56]).

Two studies evaluated EMs for UNE. One RCT compared LLLT versus continuous US [70] and the other evaluated continuous SWD plus splint versus placebo [71]. Two comparative studies evaluated EMs for the treatment of hand paralysis. One compared LLLT alone versus LLLT plus splint for radial palsy [72], and the other used ultrasound, electrostimulation, thermal and manual therapy in a unified therapeutic protocol for brachial, median, ulnar, and radial palsy [73].

**Electrophysical modalities versus placebo.** Favourable results for extracorporeal shockwave therapy plus splint in pain relief, severity of symptoms, functional status, and pinch strength. Only the result for pinch strength was supported by a moderate effect size. Conflicting evidence for low-level laser therapy; favoured in three studies and participants with mild carpal tunnel syndrome. A large effect size showed superiority of placebo over electrophysical modalities. Significant improvement in motor latency, sensory amplitude, and grip strength of low-level laser therapy plus splint (trivial effect size) and inconclusive results for sensory latency, motor amplitude, sensory, and motor conduction velocity.

Lazovic et al. [48] reported pain reduction at the end of treatment in the low-level laser therapy group and expressed the results as percentages. Arikan et al. [61] reported improvements in pain (VAS) and symptom severity in the placebo group and presented their results in ranges (min/max). We did not receive the data from the authors, so we could not include them in the meta-analysis.

We found no significant differences in the remaining modalities for the parameters mentioned. The outcomes and significance are in Table 1, and detailed analyses are presented in Figs 2–12.

**Table 1. Outcome measures and significance of electrophysical modalities versus placebo.**

| Author | Intervention | VAS | SS | FS | ML | SL | MNCV | SNCV | SNAP (A) | CMAP (A) | Grip strength | Pinch strength |
|---|---|---|---|---|---|---|---|---|---|---|---|---|
| Armagan et al. | US + SP | + | + | + | + | + | + | + | | | | |
| Jothi and Bland. | US + SP | | + | + | + | | | + | | | | |
| Oztas et al. | US | | | | + | + | + | + | | | | |
| Wu et al. | ESWT | + | + | + | | | | + | | | | + |
| Ke et al. | ESWT | | + - | + - | | | | + - | | | | |
| Abid Ali et al. | LLLT | + - | + - | + - | + - | + | + - | + | + - | + - | | |
| Jiang et al. | LLLT | + * | +—* | | + - | + - | | | | | | |
| Tascioglu et al. | LLLT | + | + | + | + | | + | + | | | + | |
| Shooshtari et al. | LLLT | + - | | | + | + | | + | | | + | |
| Chang et al. | LLLT | | + | + - | + | + | | | | | + | |
| Lazovic et al. | LLLT | + ° | | | + | | | + - | | | | |
| Fusakul et al. | LLLT + SP | + - | + - | + | + - | + - | | | + - | + | + - | + - |
| Evcik et al. | LLLT + SP | | | | | | | | | | | |
| Boyaci et al. | SWD + SP | + | + | + | + | + | | + | | | | |
| Badur et al. | SWD + SP | + | | | | | | | | | | |
| Carter et al. | SMF | + | | | | | | | | | | |
| Colbert et al. | SMF | | + | + | + | + | | | + | + | | |
| Weintraub and Cole. | SMF+PMF | + | | | | | | | | | | |
| Arikan et al. | PMF | + ° | + | | + | + - | + | + | + | + | | |

Abbreviations: VAS: Visual analog scale; SS: Symptom severity; FS: Functional status; ML: Motor latency; SL: Sensory latency; MNCV: Motor nerve conduction velocity; SNCV: Sensory nerve conduction velocity; SNAP (A): Sensory nerve action potential amplitude; CMAP (A): Compound muscle action potential amplitude. Annotation symbols: Measured: +; Statistically significant: -; Not included in the meta-analysis: °; Only in the mild group: *.

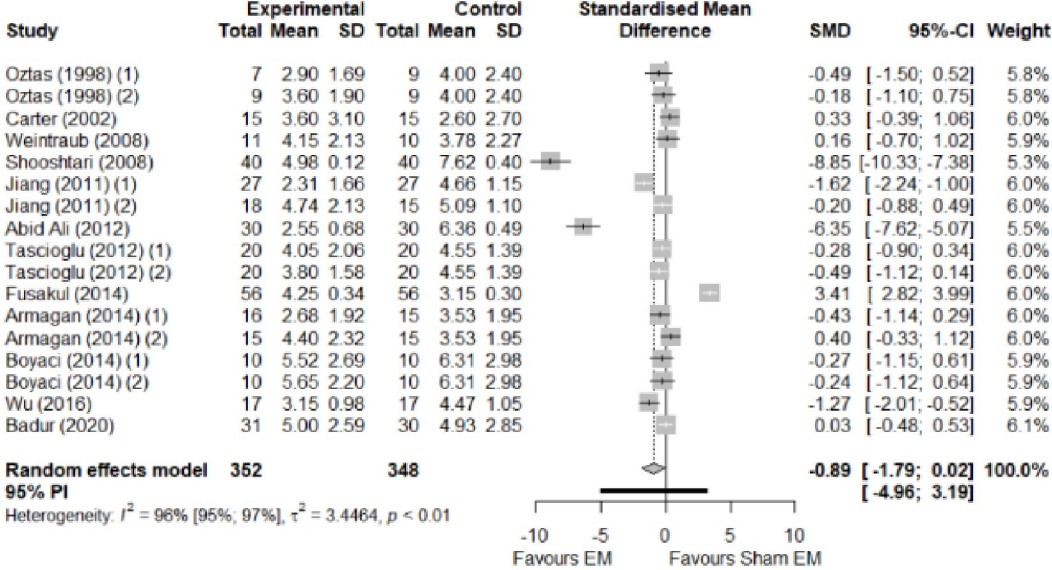

**Fig 2. Analysis—Electrophysical modalities versus placebo (pain).** Studies with more than two intervention groups (different modalities, treatment doses, or symptom classification) were numbered as (1) and (2).

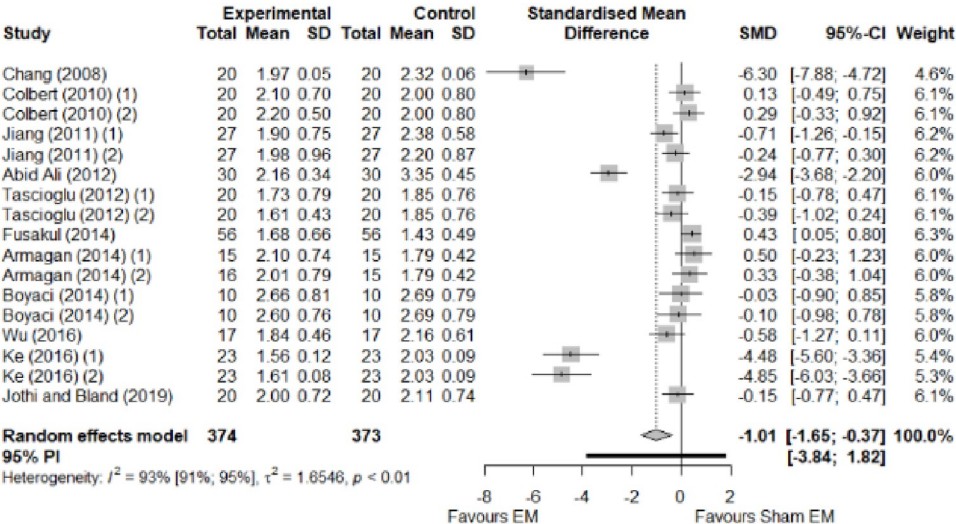

**Fig 3. Analysis—Electrophysical modalities versus placebo (symptoms severity).** Studies with more than two intervention groups (different modalities, treatment doses, or symptom classification) were numbered as (1) and (2).

**Electrophysical modalities versus manual therapy.** We observed a greater improvement (trivial effect size) in pain with low-level laser therapy versus manual therapy. We observed that fascial manipulation was superior to low-level laser therapy for symptom severity and functional status. Favourable results for low-level laser therapy in motor latency. No significant difference for low-level laser therapy plus ultrasound in neurophysiological parameters or strength. The outcomes and significance are in Table 2, and detailed analyses are presented in Figs 13–19.

**Comparison between electrophysical modalities.** We found superior results with trivial effect size for pulsed US over low-level laser therapy in pain relief, symptoms, and sensory latency for carpal tunnel syndrome and ulnar neuropathy at the elbow. Favourable results for

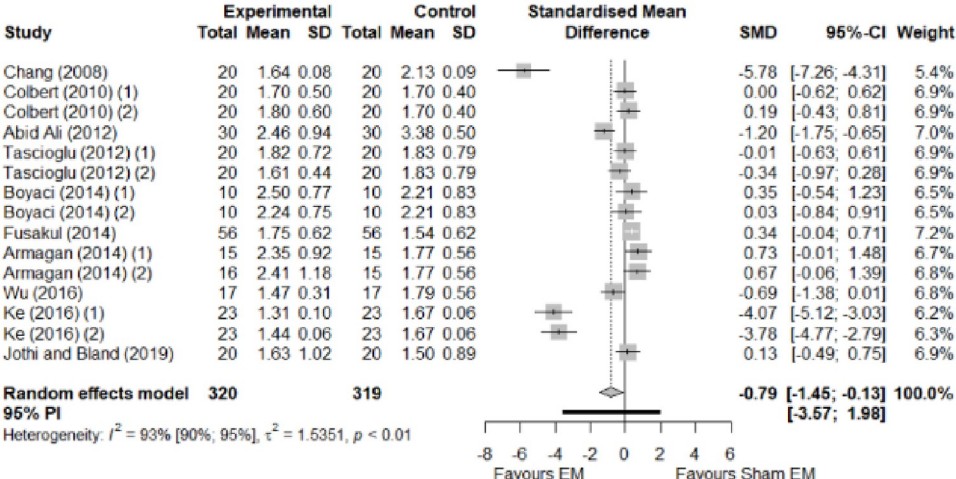

**Fig 4. Analysis—Electrophysical modalities versus placebo (functional status).** Studies with more than two intervention groups (different modalities, treatment doses, or symptom classification) were numbered as (1) and (2).

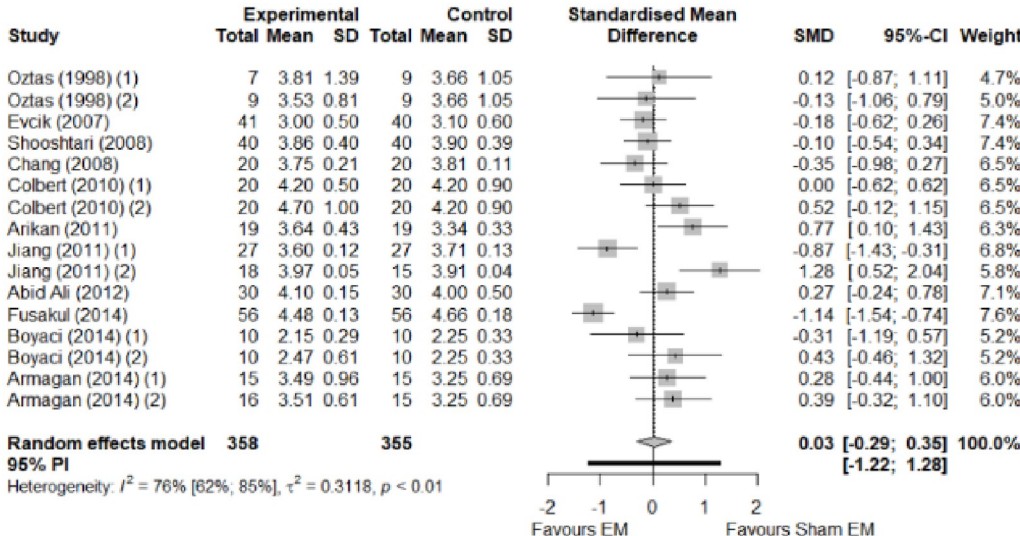

**Fig 5. Analysis—Electrophysical modalities versus placebo (sensory latency).** Studies with more than two intervention groups (different modalities, treatment doses, or symptom classification) were numbered as (1) and (2).

low-level laser therapy in motor latency and sensory velocity. Grip strength improved with both modalities of ultrasound over low-level laser therapy (large effect size). No significant difference for low-level laser therapy, transcutaneous electrical nerve stimulation, or ultrasound in the remaining parameters. The outcomes and significance are in Table 3, and detailed analyses are presented in Figs 20–26.

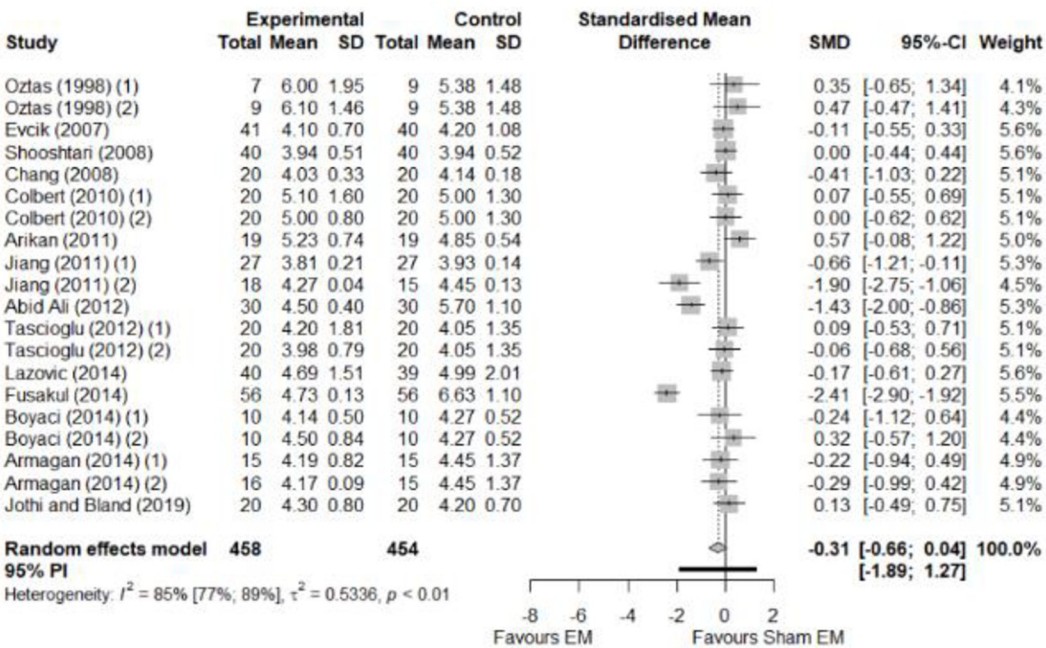

**Fig 6. Analysis—Electrophysical modalities versus placebo (motor latency).** Studies with more than two intervention groups (different modalities, treatment doses, or symptom classification) were numbered as (1) and (2).

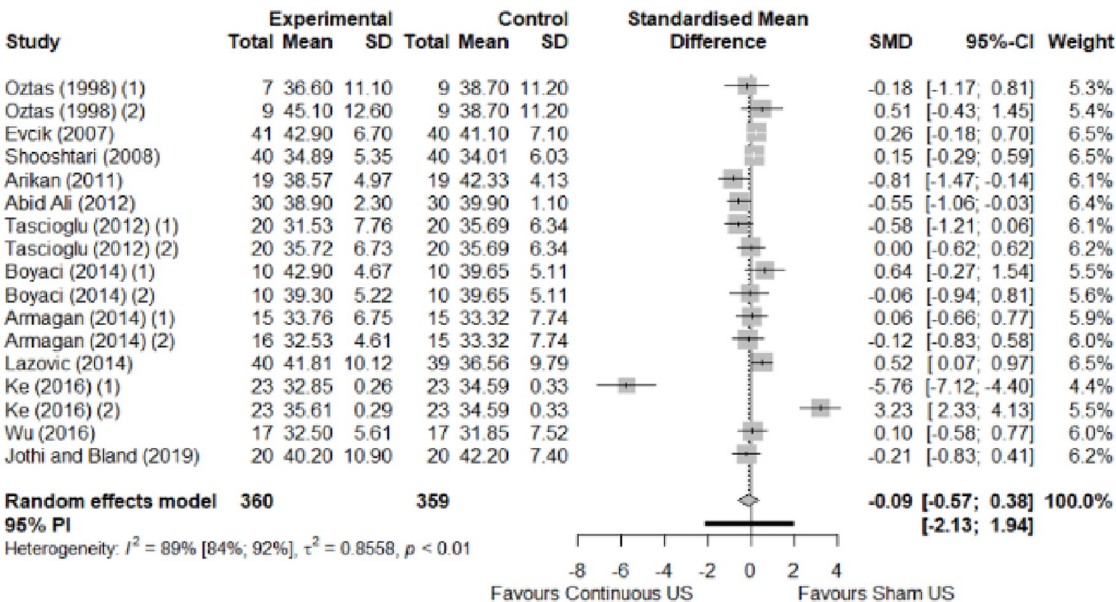

**Fig 7. Analysis—Electrophysical modalities versus placebo (sensory velocity).** Studies with more than two intervention groups (different modalities, treatment doses, or symptom classification) were numbered as (1) and (2).

Ozkan et al. [70] compared low-level laser therapy and ultrasound for ulnar neuropathy at the elbow. They reported a marked reduction in VAS pain at the end of treatment, in the first and third months of follow-up in the ultrasound group, while the low-level laser therapy group only showed improvement in the first month of follow-up. Dakowicz et al. [67] compared low-level laser therapy and pulsed magnetic field for carpal tunnel syndrome. They reported a significant reduction in VAS pain in both groups after each treatment series and six months after the last series. The authors presented their mean values through a graph. We did not receive the data from the authors, so we could not include them in the meta-analysis.

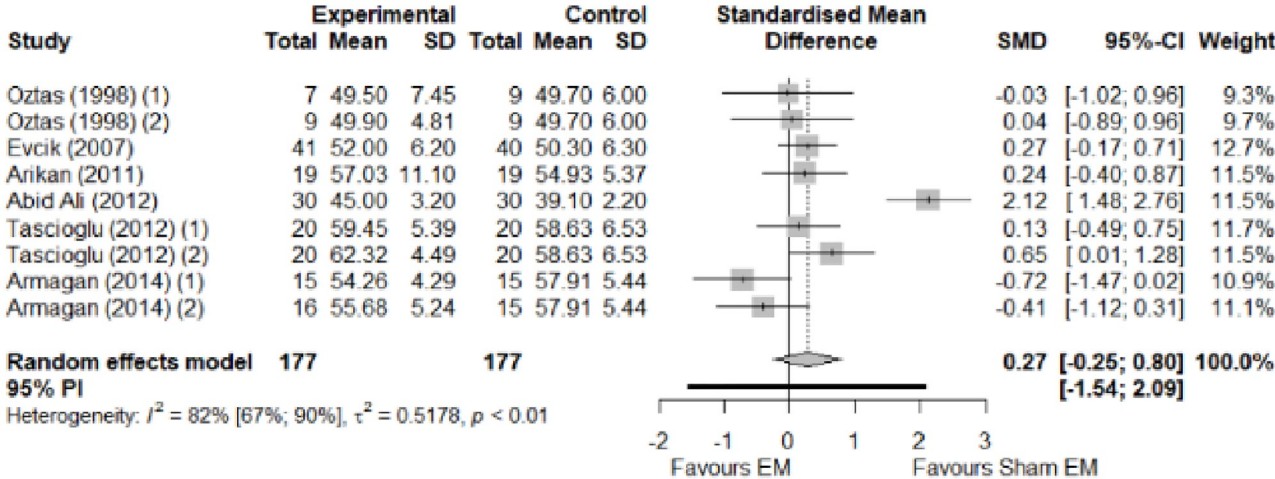

**Fig 8. Analysis—Electrophysical modalities versus placebo (motor velocity).** Studies with more than two intervention groups (different modalities, treatment doses, or symptom classification) were numbered as (1) and (2).

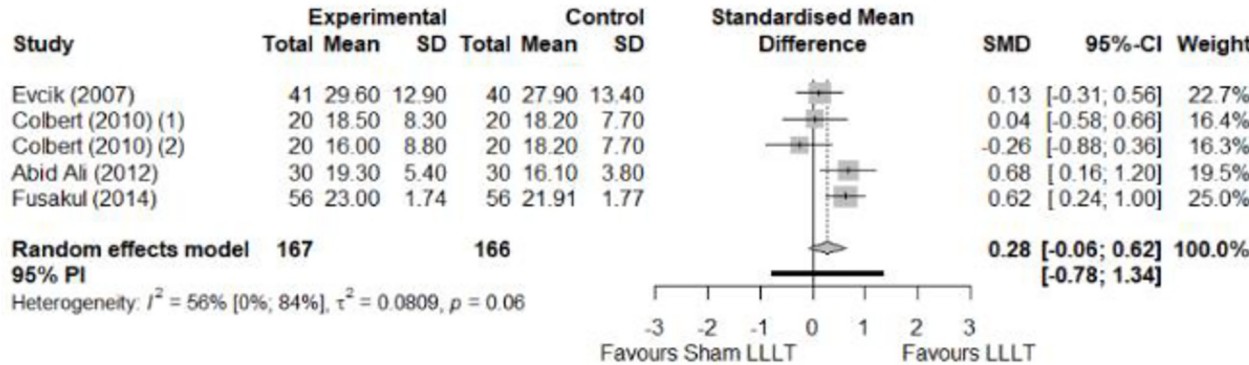

**Fig 9. Analysis—Electrophysical modalities versus placebo (sensory nerve action potential amplitude).** Studies with more than two intervention groups (different modalities, treatment doses, or symptom classification) were numbered as (1) and (2).

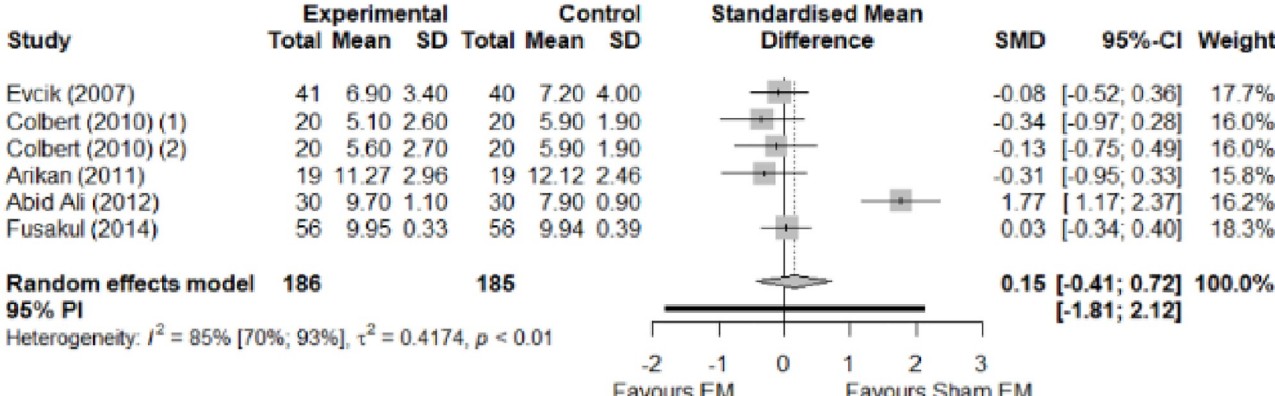

**Fig 10. Analysis—Electrophysical modalities versus placebo (compound muscle action potential amplitude).** Studies with more than two intervention groups (different modalities, treatment doses, or symptom classification) were numbered as (1) and (2).

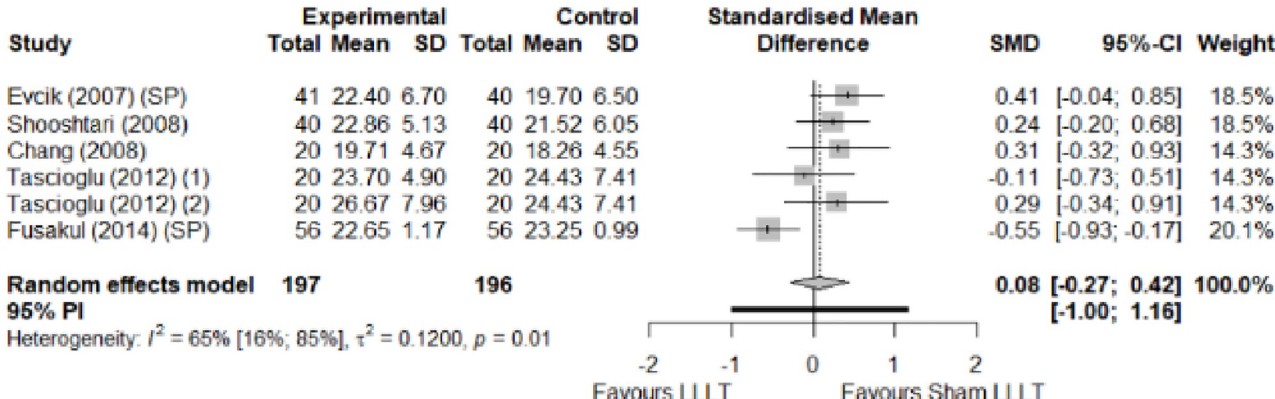

**Fig 11. Analysis—Electrophysical modalities versus placebo (grip strength).** Studies with more than two intervention groups (different modalities, treatment doses, or symptom classification) were numbered as (1) and (2). Modalities delivered with a splint were marked as (SP).

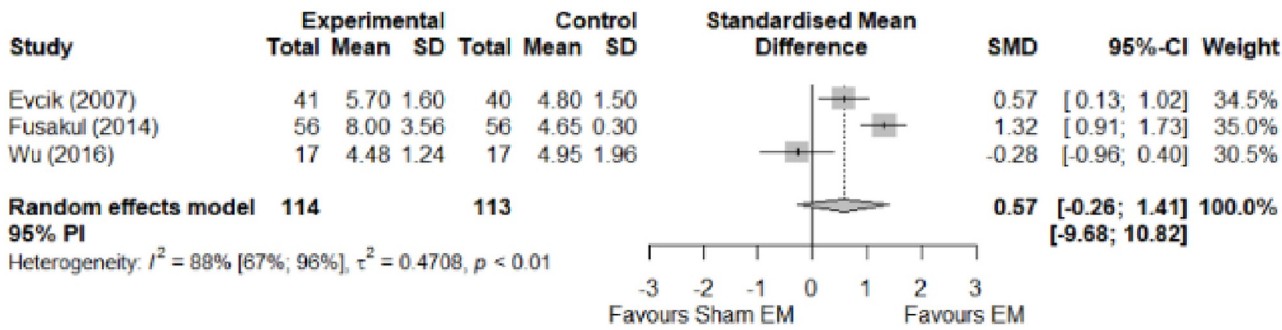

**Fig 12. Analysis—Electrophysical modalities versus placebo (pinch strength).**

**Electrophysical modalities versus splinting.** Most of the favourable results correspond to the use of splinting in conjunction with electrophysical modalities. This evidence shows a moderate effect size. We found favourable results for pain relief with pulsed Radiofrequency and low-level laser therapy plus splint and interferential current therapy alone. Significant improvement in symptom severity, functional status, sensory nerve conduction velocity, and motor latency for low-level laser therapy plus splint. No significant differences in the remaining modalities for the parameters mentioned. The outcomes and significance are in Table 4, and detailed analyses are presented in Figs 27–34.

## Clinical significance

As suggested by Lemieux et al [74] and Page [75], we calculated the MCID by multiplying the pooled baseline standard deviation values by 0.2, which corresponds to the smallest effect size.

We compare the results of the meta-analysis with the references of the minimal clinically important differences for VAS (MCID of 1.2) [76], FSS (MCID of 0.74) [77], SSS (MCID of 1.04) [78], grip strength (MCID of 2.69 kg) and pinch strength (MCID of 0.68 kg) [79] and did not find any results that could be clinically significant. The overview of MCID estimation is in Table 5.

**Table 2. Outcome measures and significance of electrophysical modalities versus manual therapy.**

| Author | INT | VAS | SS | FS | ML | SL | MNCV | SNCV | SNAP (A) | CMAP (A) | Grip strength | Pinch strength | *Muscle strength* |
|---|---|---|---|---|---|---|---|---|---|---|---|---|---|
| Atya and Mansour | LLLT* vs N/TGE | + - | | | + - | + - | | + - | | | + - | | |
| Pratelli et al. | LLLT vs FM* | + - | + - | + - | | | | | | | | | |
| Milicin & Sîrbu. | US + TT + ES + KT + MM | | | | | | + | | | | | | + |
| Baysal et al. | US + SP vs N/TGE + SP vs N/TGE + US + SP* | + | + | + | + | + | | | | | + | | |
| Wolny et al. | US + LLLT vs NDT + CBM + MM | | + | + | | + | | + | | | | | |

Abbreviations: INT: Intervention; VAS: Visual analog scale; SS: Symptom severity; FS: Functional status; ML: motor latency; SL: Sensory latency; MNCV: Motor nerve conduction velocity; SNCV: Sensory nerve conduction velocity; SNAP (A): Sensory nerve action potential amplitude; CMAP (A): Compound muscle action potential amplitude; N/TGE: Nerve and tendon gliding exercises; FM: Fascial manipulation; TT: Thermotherapy; ES: Electrostimulation; KT: Kinesiotherapy; MM: Manual massage; SP: Splint. NDT: Neurodynamic Technique; CBM: Carpal bone mobilisation. Annotation symbols: +, Measured; -, Statistically significant; *, Evidence favours this intervention.

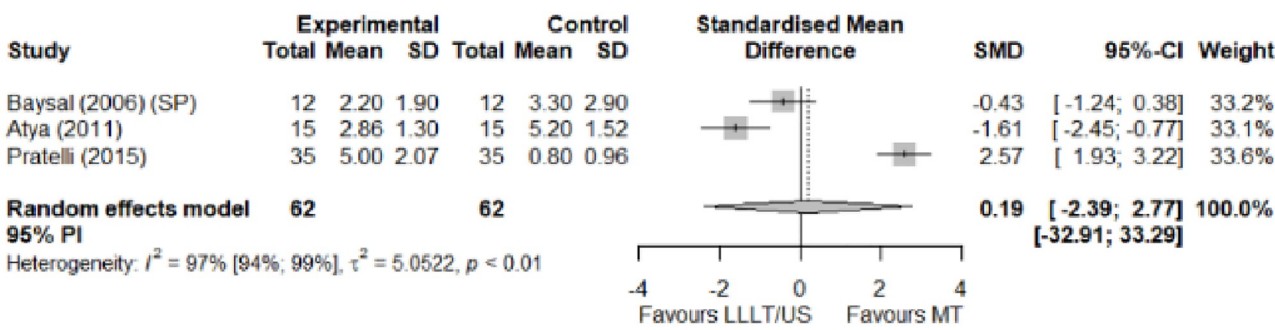

**Fig 13. Analysis—Electrophysical modalities versus manual therapy (pain).** Studies delivering modalities with a splint were marked as (SP).

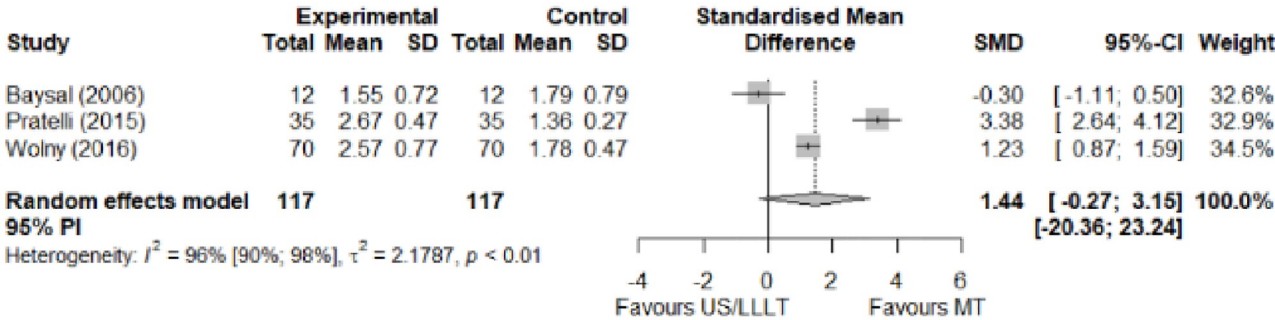

**Fig 14. Analysis—Electrophysical modalities versus manual therapy (symptoms severity).**

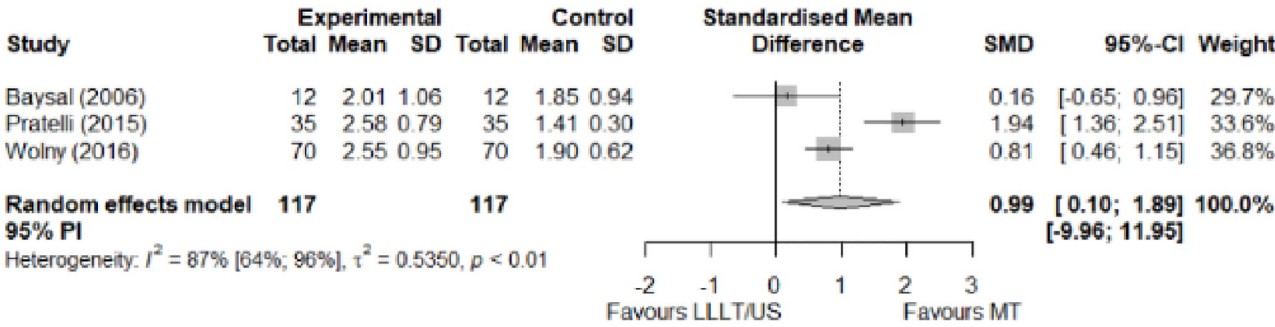

**Fig 15. Analysis—Electrophysical modalities versus manual therapy (functional status).**

**Fig 16. Analysis—Electrophysical modalities versus manual therapy (sensory latency).** Studies delivering modalities with a splint were marked as (SP).

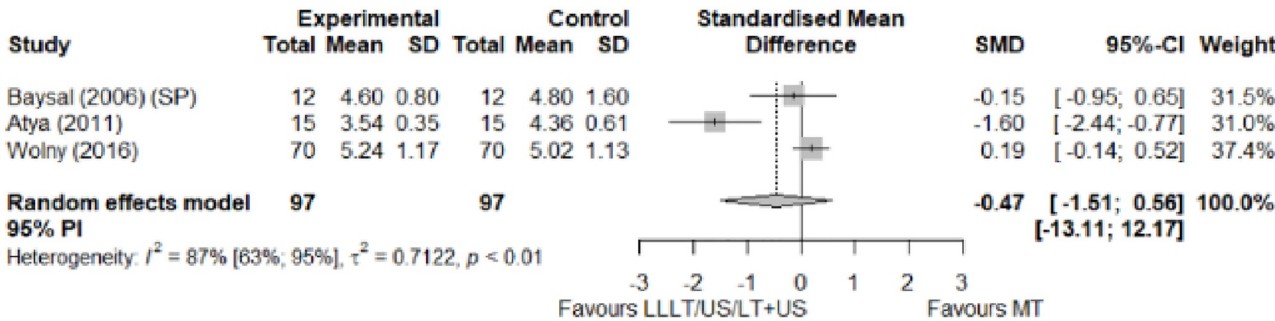

**Fig 17. Analysis—Electrophysical modalities versus manual therapy (motor latency).** Studies delivering modalities with a splint were marked as (SP).

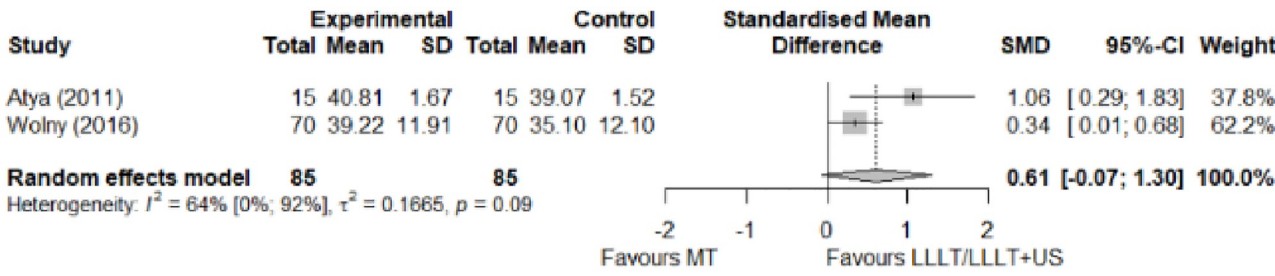

**Fig 18. Analysis—Electrophysical modalities versus manual therapy (sensory velocity).**

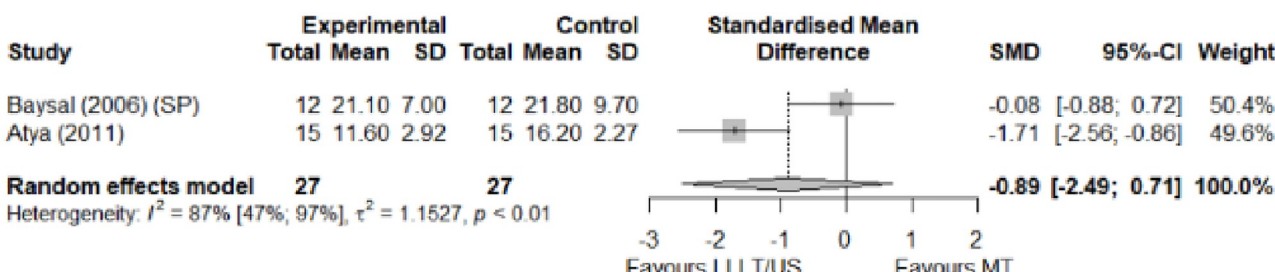

**Fig 19. Analysis—Electrophysical modalities versus manual therapy (grip strength).** Studies delivering modalities with a splint were marked as (SP).

## Discussion

This review included 38 studies comparing the effects of electrophysical modalities compared to placebo or other non-surgical (non-pharmacological) treatments to manage traumatic peripheral neuropathies. We carried out a detailed analysis assessing the main parameters associated with symptoms, function, strength, and nerve conduction.

We assessed the quality of evidence using the GRADE approach. The quality in most studies was classified as low or very low. Risk of bias varied between studies, but was generally serious in most domains. Heterogeneity was mostly high. All studies were small, ranging from 18 to 140 participants, so it is plausible that any effects may be overestimated.

The results of meta-analysis revealed that ESWT plus splint could improve symptoms and functional parameters in patients with mild or moderate carpal tunnel syndrome. Our findings are similar to the results from Huisstede et al. [80], who reported moderate evidence regarding

**Table 3. Outcome measures and significance of the comparison between electrophysical modalities.**

| Author | Intervention | VAS | SS | FS | ML | SL | MNCV | SNCV | SNAP (A) | CMAP (A) | Grip strength | Pinch strength | Muscle power |
|---|---|---|---|---|---|---|---|---|---|---|---|---|---|
| Ozkan et al. | LLLT vs Continuous US* | + | + | | | + | | | | | + | | |
| Oshima et al. | LLLT vs LLLT + SP | | | + | | | | | | | | | + |
| Armagan et al. | Pulsed US + SP vs Continuous US + SP | + - | + | + | + | + | + | | + | | | | |
| Boyaci et al. | Pulsed SWD + SP vs Continuous SWD + SP | + | + | + | + | + | | | + | | | | |
| Casale et al. | LLLT vs TENS | + | | | + - | | | | + | | | | |
| Dakowicz et al. | LLLT vs PMF | + ° | | | | | | | | | | | |
| Saeed et al. | LLLT vs Pulsed US | + - | + - | + | + + | + - | | | | | | | |
| Dincer et al. | LLLT+ SP vs Continuous US + SP | + | + | + | + | | | | + | | | | |
| Paoloni et al. | ESWT vs Pulsed US vs Pulsed Cryo US | + | + | + | | | | | | | | | |
| Koca et al. | TENS vs IFC | + - | + | + | + | | | | + | | | | |

Abbreviations: VAS: Visual analog scale; SS: Symptom severity; FS: Functional status; ML: motor latency; SL: Sensory latency; MNCV: Motor nerve conduction velocity; SNCV: Sensory nerve conduction velocity; SNAP (A): Sensory nerve action potential amplitude; CMAP (A): Compound muscle action potential amplitude; SP: Splint. *Annotation symbols*: +, Measured; -, Statistically significant; *, Evidence favours this intervention; °, Not included in the meta-analysis.

the effectiveness of radial ESWT compared with placebo ESWT in the short-term. We concur with the results from Kim et al. [81], who noticed effectiveness in the outcomes mentioned above but differed in the electrophysiological parameters' findings.

The US appears to be more effective than LLLT in improving grip strength, pain, and sensory latency. However, we found no significant differences compared to placebo or manual therapy in line with the results of Page et al. [25], who found effectiveness in the outcomes mentioned above but differed in the electrophysiological findings. We also agree with the authors, who noted there is no evidence that US applied with a splint is more effective than any other non-surgical intervention.

Similar to Huisstede et al. [80], we found limited evidence (from one RCT) that fascial manipulation can improve functional and symptom outcomes. Likewise, LLLT plus splint and PRF plus splint compared to splinting. Besides, our results showed that LLLT plus splint was superior to placebo in terms of improving grip strength in patients with mild to moderate CTS, confirming the findings of Bekhet et al. [23] and Li et al. [28]. For the rest of the parameters, we found conflicting evidence differing from the results obtained by Li et al. [28] and is

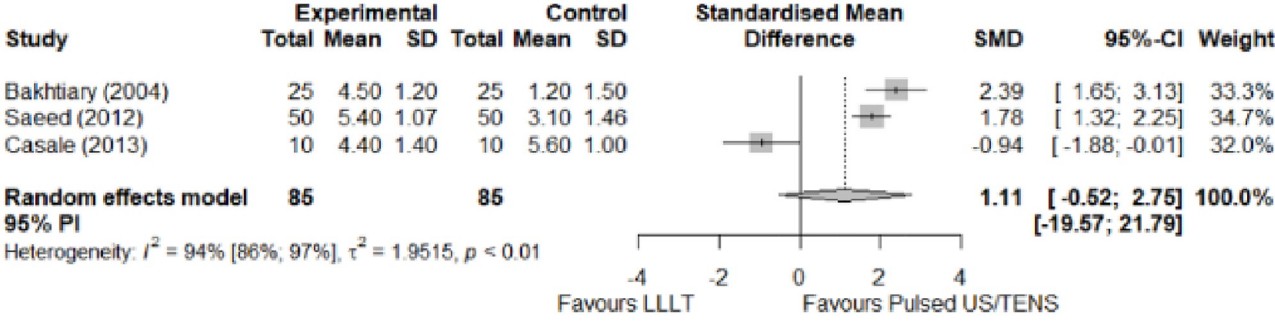

**Fig 20. Analysis—Comparison between electrophysical modalities (pain).**

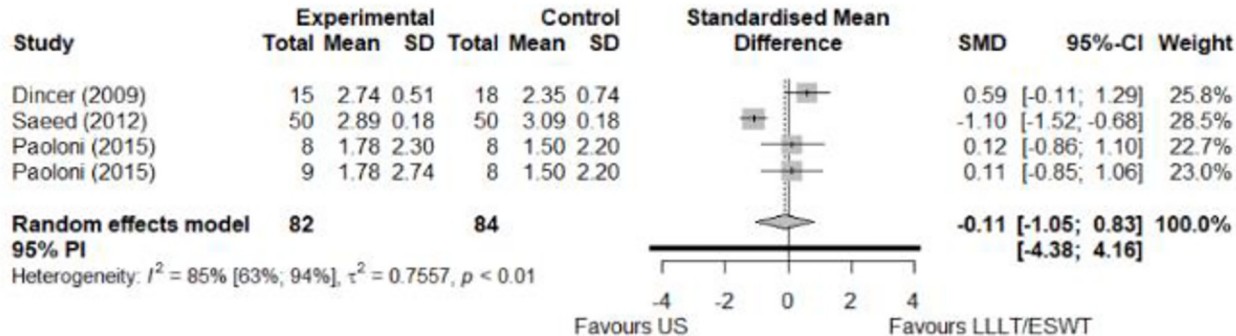

**Fig 21. Analysis—Comparison between electrophysical modalities (symptoms severity).**

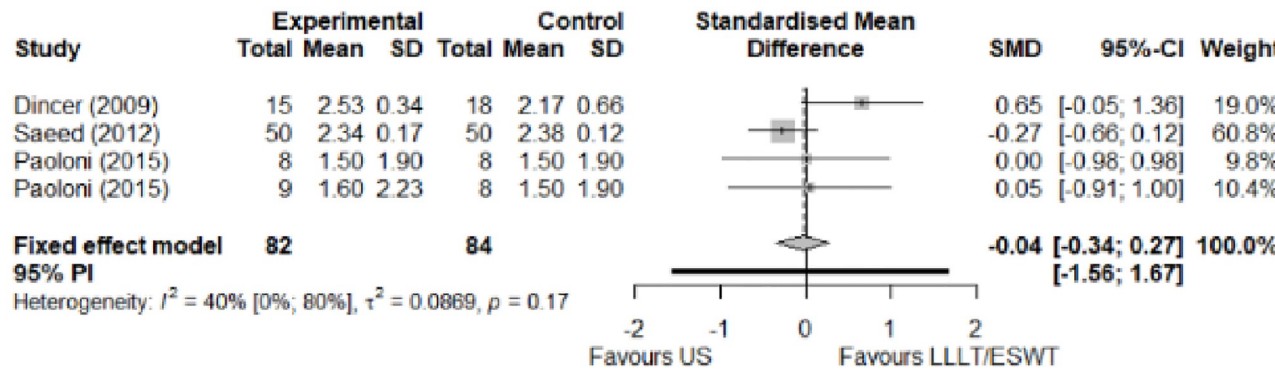

**Fig 22. Analysis—Comparison between electrophysical modalities (functional status).**

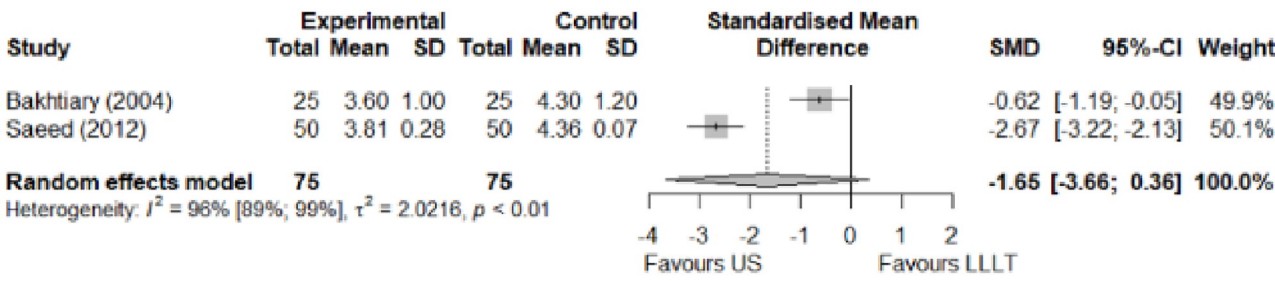

**Fig 23. Analysis—Comparison between electrophysical modalities (sensory latency).**

consistent with those obtained by Burger et al. [30]. We agree with the observation made by Bekhet et al. [23] and Li et al. [28], highlighting the usefulness of orthoses as an agent of influencing the outcomes of peripheral neuropathies.

We found no evidence for the effectiveness of magnetic field therapy in functional and symptom improvement or electrophysiological parameters. Our findings agree with O'Connor et al. [29] and differ from Huisstede et al. [80], who reported conflicting evidence.

We found no evidence of the effectiveness of SWD, PPNL, or TENS. Our results differ from Huisstede et al. [80] for SWD and are similar concerning PPNL. We agree with Gibson et al. [82] regarding TENS. The overview of evidence is in Table 6.

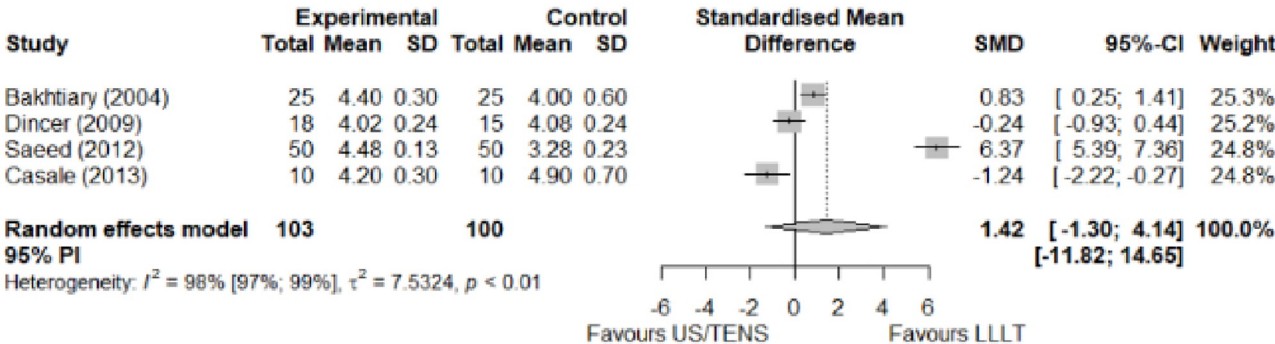

**Fig 24. Analysis—Ccomparison between electrophysical modalities (motor latency).**

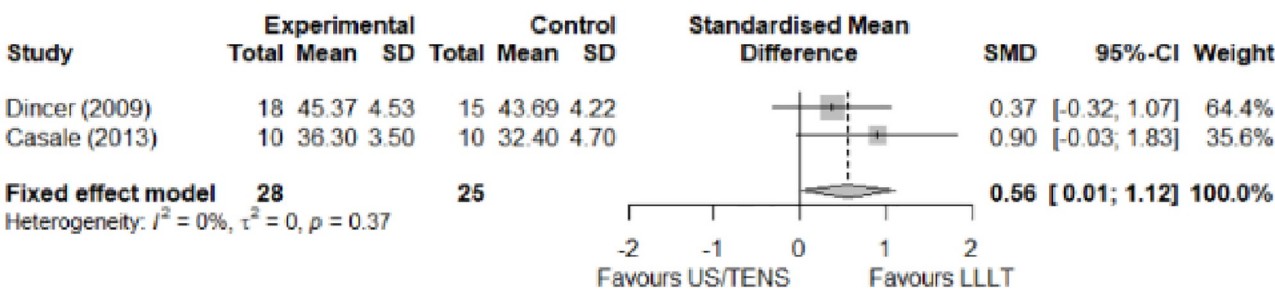

**Fig 25. Analysis—Comparison between electrophysical modalities (sensory velocity).**

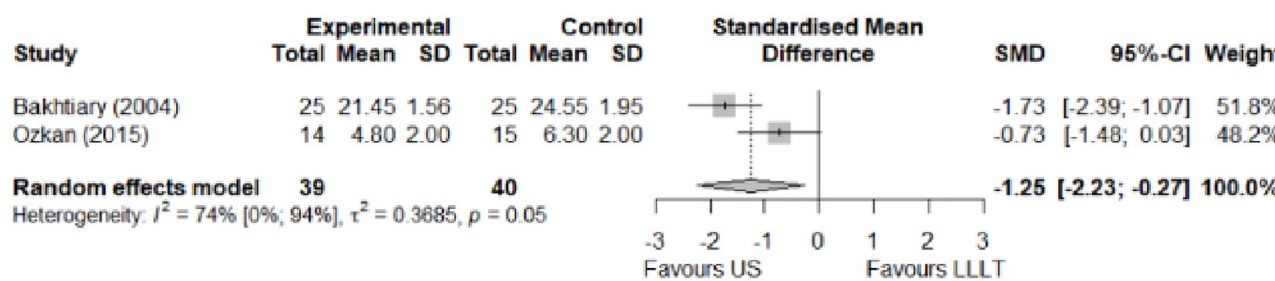

**Fig 26. Analysis—Comparison between electrophysical modalities (grip strength).**

**Table 4. Outcome measures and significance of electrophysical modalities versus splinting.**

| Author | INT | VAS | SS | FS | ML | SL | MNCV | SNCV | SNAP (A) | CMAP (A) | Grip strength | Pinch strength |
|---|---|---|---|---|---|---|---|---|---|---|---|---|
| Dincer et al. | LLLT + SP vs SP | + - | + - | + - | + - | | | + - | | | | |
| Koca et al. | TENS vs IFC vs SP | + - * | + | + | + | | | + - * | | | | |
| Chen et al. | PRF + SP vs SP | + - | + - | + - | | | | + | | | | + |
| Raeissadat et al. | PPNL+ SP vs SP | + | + ° | + ° | + ° | + | | | + ° | + ° | | |
| Raissi et al. | ESWT + SP vs SP | + | | + ° | | | | | + | + - | | |
| Yagci et al. | LLLT + SP vs SP | | + | + | + | | | + | | | | + |

Abbreviations: INT: Intervention; VAS: Visual analog scale; SS: Symptom severity; FS: Functional status; ML: motor latency; SL: Sensory latency; MNCV: Motor nerve conduction velocity; SNCV: Sensory nerve conduction velocity; SNAP (A): Sensory nerve action potential amplitude; CMAP (A): Compound muscle action potential amplitude. SP: Splint. Annotation symbols: +, Measured; -, Statistically significant; *, Evidence favours this intervention; °, Not included in the meta-analysis.

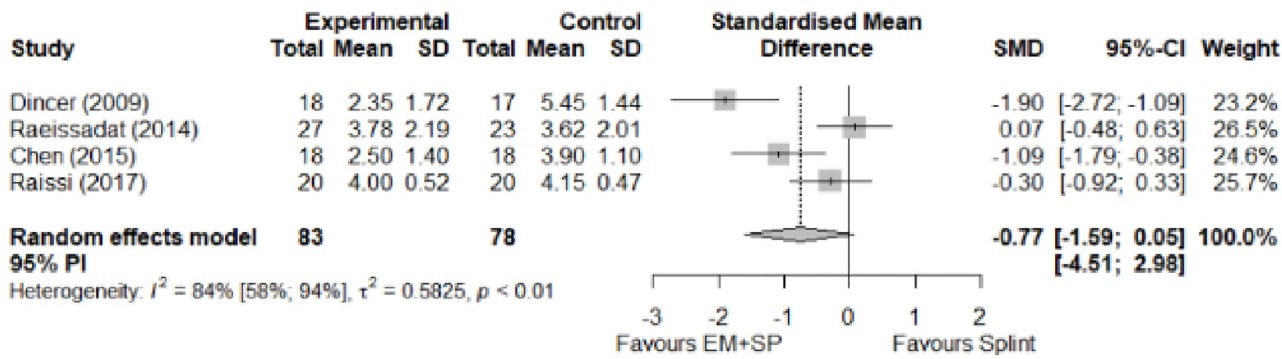

**Fig 27. Analysis—Electrophysical modalities plus splint versus splinting (pain).**

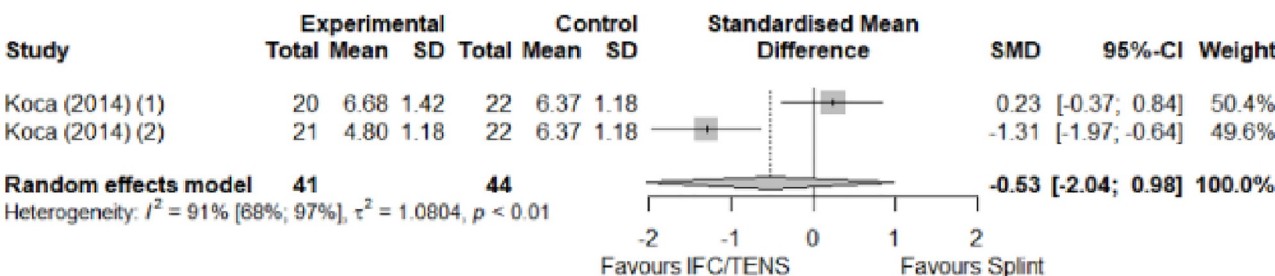

**Fig 28. Analysis—Electrophysical modalities alone versus splinting (pain).** Studies with more than two intervention groups (different modalities) were numbered as (1) and (2).

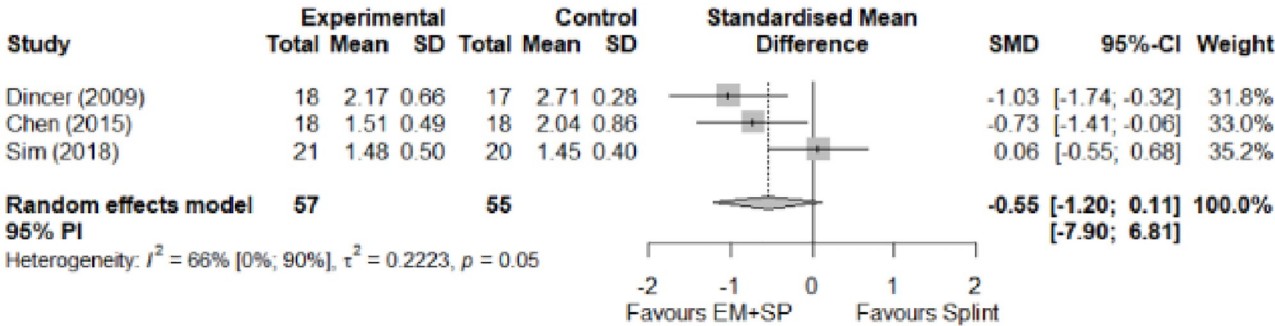

**Fig 29. Analysis—Electrophysical modalities versus splinting (symptoms severity).**

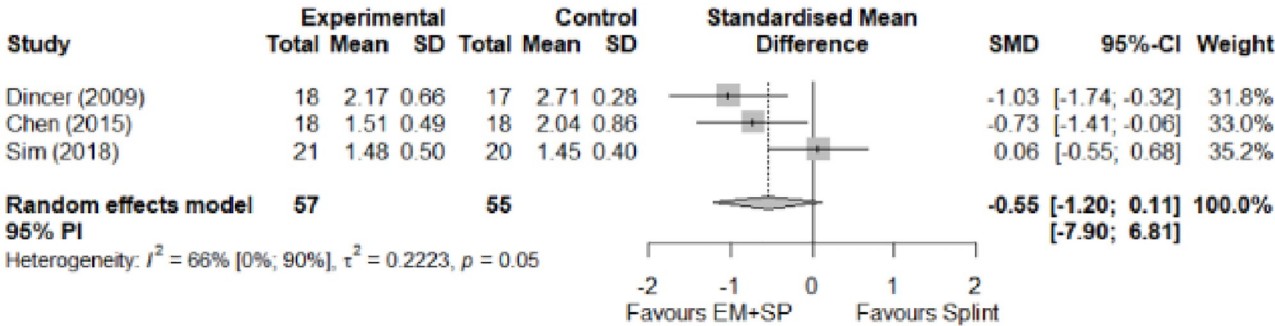

**Fig 30. Analysis—Electrophysical modalities versus splinting (functional status).**

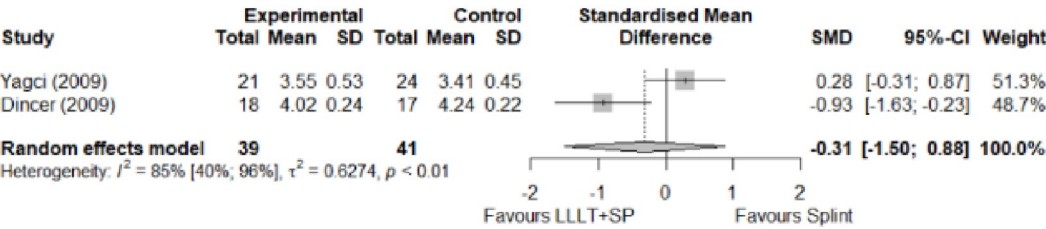

**Fig 31. Analysis—Low-level laser plus splint versus splinting (motor latency).**

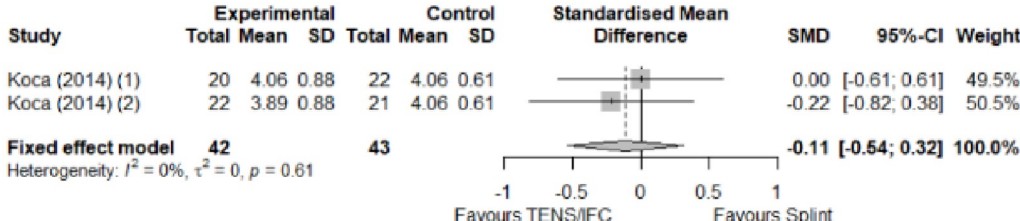

**Fig 32. Analysis—Electrophysical modalities alone versus splinting (motor latency).** Studies with more than two intervention groups (different modalities) were numbered as (1) and (2).

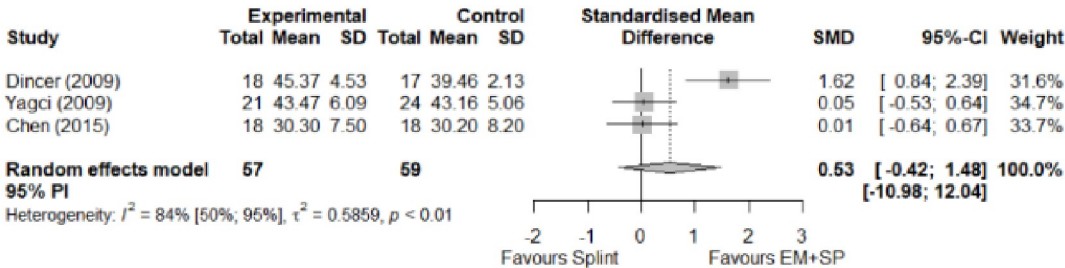

**Fig 33. Analysis—Electrophysical modalities plus splint versus splinting (sensory velocity).**

By contrasting effect sizes we could identify that the results favouring placebo were supported by large (for pain and symptom severity) and moderate (functional status) effect sizes. The only outcomes in favour of electrophysical modalities supported by a large effect size were associated with improvement in symptom severity and functional status in comparison to

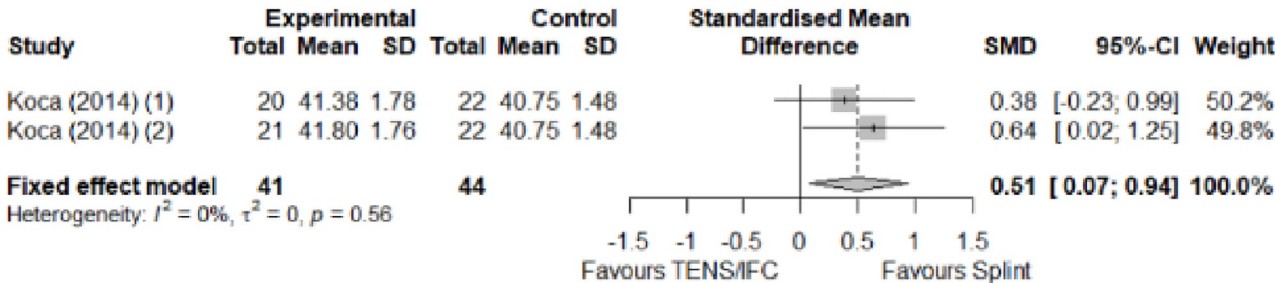

**Fig 34. Electrophysical modalities versus splinting (sensory velocity).** Studies with more than two intervention groups (different modalities) were numbered as (1) and (2).

**Table 5. Clinical significance from MCID estimation.**

| Type of comparison | Outcome | SMD [95%-CI] | *Effect size | Pooled baseline SD | Estimated MCID | Reference MCID |
|---|---|---|---|---|---|---|
| **Electrophysical modalities vs placebo** | Pain (VAS) | -0.89 [-1.79; 0.02] | Large | 1.6 | 0.32 | 1.2 |
| | Symptom Severity (SSS) | -1.01 [-1.65; -0.37] | Large | 0.55 | 0.11 | 1.04 |
| | Functional Status (FSS) | -0.79 [-1.45; -0.13] | Moderate | 0.63 | 0.13 | 0.74 |
| | Grip Strength | 0.08 [-0.22; 0.37] | Trivial | 5.4 | 1.08 kg | 2.69 kg |
| | Pinch Strength | 0.57 [-0.26; 1.41] | Moderate | 1 | 0.20 kg | 0.68 kg |
| **Electrophysical modalities vs Manual therapy** | Pain (VAS) | 0.19 [-2.39; 2.77] | Trivial | 2.37 | 0.47 | 1.2 |
| | Symptom Severity (SSS) | 1.44 [-0.27; 3.15] | Large | 4.06 | 0.81 | 1.04 |
| | Functional Status (FSS) | 0.99 [0.10; 1.89] | Large | 0.95 | 0.19 | 0.74 |
| | Grip Strength | -0.89 [-2.49; 0.71] | Large | 6.11 | 1.22 kg | 2.69 kg |
| **Electrophysical modalities vs Splinting** | Pain (VAS) | -0.77 [-1.59; 0.05] | Moderate | 1.35 | 0.27 | 1.2 |
| | Symptom Severity (SSS) | -0.66 [-1.33; 0.01] | Moderate | 2.84 | 0.57 | 1.04 |
| | Functional Status (FSS) | -0.55 [-1.20; 0.11] | Moderate | 0.77 | 0.15 | 0.74 |
| **Comparison between electrophysical modalities** | Pain (VAS) LLLT vs Other EM | 1.11 [-0.52; 2.75] | Large | 2.9 | 0.58 | 1.2 |
| | Symptom Severity (SSS)–US vs Other EM | -0.11 [-1.05; 0.83] | Trivial | 1.39 | 0.28 | 1.04 |
| | Functional Status (FSS)–US vs Other EM | -0,04 [-0.34; 0,27] | Trivial | 1.16 | 0.25 | 0.74 |
| | Grip Strength–LLLT vs US | -1.25 [-2.23; -0.27] | Large | 6.1 | 1.22 kg | 2.69 kg |

*Cohen's d coefficient: <0.2 = trivial effect; 0.2–0.5 = small effect; 0.5–0.8 = moderate effect; > 0.8 = large effect.

manual therapy. The superior results of splinting over electrophysical modalities were supported by moderate effect sizes. Likewise a moderate effect favoured electrophysical modalities over placebo in pinch strength. The results favouring ultrasound over the other modalities were supported by trivial effect sizes. Similarly a trivial effect size was associated with grip strength in favour of modalities over placebo and in pain improvement over manual therapy.

We contrasted the results with the minimal clinically important difference (MCID) in order to provide practical evidence to support clinical decision-making in the use of therapeutic alternatives for the management of peripheral neuropathies. We found no clinical significance in any of the pooled results when compared to the MCID.

## Strengths and limitations

To our knowledge, this is the first systematic review of the effectiveness of electrophysical modalities to treat traumatic neuropathies of the wrist and hand. We used the protocols and methodological tools that ensured quality and transparency in selecting, screening, and treating data.

**Table 6. Overview of evidence of electrophysical modalities.**

| Low-level laser therapy | ST | MT | Ultrasound | ST | MT | Magnetic field therapy | ST | MT |
|---|---|---|---|---|---|---|---|---|
| LLLT alone vs placebo | ± | | Continuous US splint vs Pulsed US plus splint vs placebo plus splint | Ø | | Pulsed magnetic field vs placebo | + | |
| LLLT plus splint* vs placebo plus splint | + | | Continuous US ($1.5W/cm^2$ * vs $0.8W/cm^2$ dosage) vs placebo | + | | Static magnetic field vs placebo | Ø | |
| LLLT vs Pulsed Magnetic Field ° | Ø | Ø | Pulsed US plus splint vs MT plus splint vs pulsed US plus MT plus splint* | + | | Static magnetic field (15mT vs 45mT dosage) vs placebo | Ø | |
| LLLT* vs TENS | + | | Pulsed US vs Cryo-US vs ESWT* | Ø | Ø | Static + pulsed magnetic field | Ø | |
| LLLT vs Continuous ultrasound* | + | | Pulsed US* vs LLLT | + | | **Polarised polychromatic non-coherent light** | | |
| LLLT plus splint vs Continuous US plus splint | Ø | | Continuous US plus splint* vs splinting | + | | PPNL (Bioptron) plus splint vs splinting | Ø | |
| LLLT alone vs LLLT plus splint° | Ø | | **Extracorporeal short-wave therapy** | | | **Short-wave diathermy** | | |
| LLLT plus splint* vs splinting | + | Ø | ESWT plus splint* vs placebo plus splint | + | Ø | Continuous SWD vs Pulsed SWD vs placebo SWD | Ø | |
| LLLT vs Fascial Manipulation* | + | Ø | ESWT plus splint* vs splinting | + | | **Pulsed radiofrequency** | | |
| LLLT* vs Nerve and Tendon gliding exercises | + | | ESWT (2 dosages) vs placebo | + | + | Pulsed radiofrequency plus splint* vs splinting | + | + |
| | | | **Interferential current** | | | | | |
| | | | Interferential current* vs TENS vs split | + | | | | |

Abbreviations: ST, Short-term; MT, Mid-term; ±, Conflicting evidence; +, Limited evidence; *, Evidence favours this intervention; Ø, No difference; °, Not included in the meta-analysis.

One of the main limitations to have a broader picture of all pathologies was the scarce availability of studies evaluating traumatic peripheral neuropathies. We found a predominance of trials studying entrapment injuries (94.7% of these trials assessed CTS), and only two trials assessed hand paralysis [72,73]. We do not include studies published in a language other than English.

## Conclusions

### Implications for practice

We found favourable results for ESWT and PRF in pain relief, symptom severity, functional status, sensory conduction velocity, motor latency, and motor amplitude in participants with CTS. Conflicting evidence of the effectiveness of LLLT for FSS and neurophysiological parameters in participants with mild to moderate CTS.

Continuous US was superior to LLLT in pain and symptom relief in participants with UNE. We found no evidence of benefit in other modalities and parameters.

Although we found some differences favouring electrophysical modalities, mainly when applied with a splint, none of the results obtained throughout this review can be considered clinically significant.

### Implications for research

This review found mainly RCTs assessing entrapment injuries with the prevalence of CTS. More high-quality research is needed to evaluate the effectiveness of the available treatments for brachial, radial, ulnar, and median neuropathies, including those with more considerable complexity and rehabilitation time, such as axonotmesis.

## Supporting information

**S1 Table. PRISMA checklist.**
(DOC)

**S2 Table. PICO question.**
(DOCX)

**S3 Table. GRADE summary of findings.**
(PDF)

**S4 Table. Risk of bias of randomised controlled studies.**
(XLSX)

**S5 Table. Measures and outcomes of included studies.**
(DOCX)

**S1 File. PROSPERO protocol.**
(PDF)

**S2 File. Search terms.**
(DOCX)

## Author Contributions

**Conceptualization:** Ena Bula-Oyola, Juan-Manuel Belda-Lois, Rosa Porcar-Seder.

**Data curation:** Ena Bula-Oyola, Juan-Manuel Belda-Lois.

**Formal analysis:** Juan-Manuel Belda-Lois, Rosa Porcar-Seder.

**Investigation:** Ena Bula-Oyola, Juan-Manuel Belda-Lois, Rosa Porcar-Seder.

**Methodology:** Ena Bula-Oyola, Juan-Manuel Belda-Lois, Rosa Porcar-Seder.

**Supervision:** Juan-Manuel Belda-Lois, Rosa Porcar-Seder, Álvaro Page.

**Validation:** Juan-Manuel Belda-Lois, Rosa Porcar-Seder, Álvaro Page.

**Writing – original draft:** Ena Bula-Oyola.

**Writing – review & editing:** Ena Bula-Oyola, Álvaro Page.

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
