## [Decision Letter · Decision Letter 0]

30 Sep 2020

PONE-D-20-21315

Effectiveness of electrophysical modalities in the sensorimotor rehabilitation of radial, ulnar, and median neuropathies: a meta-analysis

PLOS ONE

Dear Dr. Bula-Oyola,

Thank you for submitting your manuscript to PLOS ONE. After careful consideration, we feel that it has merit but does not fully meet PLOS ONE’s publication criteria as it currently stands. Therefore, we invite you to submit a revised version of the manuscript that addresses the points raised during the review process.

We look forward to receiving your revised manuscript.

Kind regards,

Leila Harhaus

Academic Editor

PLOS ONE

2. We note you have included tables to which you do not refer in the text of your manuscript. Please ensure that you refer to Tables 1, 3, 4, 5, 6, 7 in your text; if accepted, production will need this reference to link the reader to the Tables.

Reviewers' comments:

Reviewer's Responses to Questions

**Comments to the Author**

1. Is the manuscript technically sound, and do the data support the conclusions?

Reviewer #1: Partly

Reviewer #2: Partly

2. Has the statistical analysis been performed appropriately and rigorously? 

Reviewer #1: I Don't Know

Reviewer #2: Yes

3. Have the authors made all data underlying the findings in their manuscript fully available?

Reviewer #1: Yes

Reviewer #2: Yes

4. Is the manuscript presented in an intelligible fashion and written in standard English?

Reviewer #1: Yes

Reviewer #2: No

5. Review Comments to the Author

Reviewer #1: Thank you for the invitation to review this paper. Our author colleagues have submitted an interesting systematic review with meta-analysis on the efficacy of electrophysical modalities in the rehabilitation of radial, ulnar, and median neuropathies. The paper concludes that there is limited favourable evidence for efficacy of electrophysical modalities in these conditions.

The authors should be commended for the comprehensive and detailed review they have undertaken.

I have some comments for consideration:

Introduction:

1) Opening sentence: ‘Peripheral neuropathies are common injuries’ I suggest use pathologies rather than injury, in case of insidious onset peripheral neuropathies not attributable to trauma.

2) Paragraph 2, sentences 1 and 2: Citations should be provided for these statements ‘Current literature has focused on the efficacy of surgical and pharmacological treatments. Regarding conservative treatments, most studies evaluated the effects of electrophysical modalities in carpal tunnel syndrome.’

Methods:

3) Search strategy – Search was undertaken in April-Jul 2019. This was a while ago, re-running and updating the search should be considered.

4) Search strategy – what is the rationale for using just 2 databases? Also, what was the rationale for limiting it to 2009?

5) Study selection and data extraction – indicate which authors (by initials) did the screening, and same for data extraction. Were disagreements solved by consensus discussion?

6) Study selection and data extraction – were corresponding authors of papers contacted if data were missing or presented in graphs only?

7) Sections of the Method that are included in the Prisma guidelines are missing – the details of the methods used for summary measures, synthesis of results (e.g. meta-analysis, best evidence synthesis) and risk of bias across studies are not described. Looking at the results, meta-analysis to calculate SMD has been performed, and there is mention of limited and moderate evidence. But these have not been defined or described in the method.

8) In line with comment 7 above, meta-analysis was undertaken but there does not appear to be an assessment of the risk of bias across studies or overall confidence of the evidence (e.g. the GRADE approach). This should be done for each meta-analysis comparison to provide insight on the quality of meta-analysis evidence and confidence in the findings.

Results:

9) Paragraph 1 – Science direct and research gate are mentioned here, but were not mentioned in the Method. Were these used as additional databases for the search?

10) Reporting of the Results would be much clearer if it were arranged using the sections defined in the Prisma checklist. For example, the section labelled Evidence Hierarchy appears to detail the Study Characteristics. Risk of Bias of individual studies is currently reported in the Method section, so should be presented in the Results instead. Splitting the reporting of results into sections detailing results of individual studies, then detailing the synthesis of results (i.e. meta-analysis and qualitative synthesis) in accordance with the Prisma guidelines would be a much neater way for arranging the Results.

11) The paper is largely well-written, however there are a few grammatical errors (e.g. LLLT is defined twice on page 6), so I suggest reviewing the manuscript for such issues.

12) Forest plots – suggest adding labels to the X axis to show which direction favours the treatment and which direction favours controls

13) Meta-analysis text results – suggest including I-squared when reporting the SMD meta-analysis results, to show heterogeneity without needing to refer to the supplementary material.

Discussion:

14) The discussion is largely a summary of the results. It would be strengthen the paper to discuss how these findings relate to other studies, consider clinical implications etc.

Reviewer #2: The authors provide a systematic analyses of electrophysical therapy in periveral nerve entrapment injuries. They included all papers fom the course of the last decade and conclude a lack of evidence about the efficacy of electrophysical therapy.

Prisma guidlines were used to prepare the manuscript. I think it is an interesting topic to evaluate non surgical treatment of peripheral nerve entrapments. Unfortunately the manuscript is very exhausting to read. The authors sum up every included study with a lack of detailed discussion (for example is splinting a viable option, etc.) Preclinical data about the efficacy is not discussed and compared to clinical experience.

There is a lot of information included in an enumerated fashion.

I would suggest to rewrite the manuscript to catch the reader's attention more and meet the high standard of the journal

6. PLOS authors have the option to publish the peer review history of their article (what does this mean?). If published, this will include your full peer review and any attached files.

Reviewer #1: No

Reviewer #2: No

---

## [Author Response · Author response to Decision Letter 0]

11 Dec 2020

Dear editor and reviewers:

Thank you for the opportunity to review our manuscript. We appreciate your accurate comments, which provided useful insights to improve our review. We present our responses in bold after every comment and highlight the changes in the manuscript in blue. We consider that we have further enhanced the clarity and lightness of our manuscript. We hope that this revised version reflects all the comments and is better suited to your journal for publication.

Reviewer #1: Thank you for the invitation to review this paper. Our author colleagues have submitted an interesting systematic review with meta-analysis on the efficacy of electrophysical modalities in the rehabilitation of radial, ulnar, and median neuropathies. The paper concludes that there is limited favourable evidence for efficacy of electrophysical modalities in these conditions.

The authors should be commended for the comprehensive and detailed review they have undertaken.

We are grateful for the reviewer's insightful comments. We made every effort to address the reviewer's suggestions throughout the manuscript.

I have some comments for consideration:

Introduction: 

1. Opening sentence: ‘Peripheral neuropathies are common injuries’ I suggest use pathologies rather than injury, in case of insidious onset peripheral neuropathies not attributable to trauma. 

We replace the term ‘lesions’ with ‘pathologies’ in the Introduction’s opening sentence, on page 2, line 49. 

2. Paragraph 2, sentences 1 and 2: Citations should be provided for these statements ‘Current literature has focused on the efficacy of surgical and pharmacological treatments. Regarding conservative treatments, most studies evaluated the effects of electrophysical modalities in carpal tunnel syndrome.’

We provided the missing citations for the statements on page 3, lines 62 and 63.

Methods:

3. Search strategy – Search was undertaken in April-Jul 2019. This was a while ago, re-running and updating the search should be considered. 

We reactivate the search on 1/10/2020 and adjust the margin until 31/12/2020. 

4. Search strategy – what is the rationale for using just 2 databases? Also, what was the rationale for limiting it to 2009?

Initially, we chose to focus our review on the last decade’s findings in the two databases with the largest number of clinical trials. However, in order to have a broader perspective, we expanded the search to the following databases: Biomed Central, Ebscohost, Lilacs, Ovid, Pedro, Sage, Scopus, Science Direct, Semantic Scholar, Taylor & Francis, and Web of Science. We also adjust the period of publication from 1980 to 2020. The search strategy is on page 4. We provide an example of the search terms in S2 File. 

5. Study selection and data extraction – indicate which authors (by initials) did the screening, and same for data extraction. Were disagreements solved by consensus discussion?

As suggested, we include the initials of the authors who did the process. Discrepancies were resolved through discussion and consensus moderated by a third author. Data section, pages 4 and 5.

6. Study selection and data extraction – were corresponding authors of papers contacted if data were missing or presented in graphs only?

We contacted the authors but did not respond.

7. Sections of the Method that are included in the Prisma guidelines are missing – the details of the methods used for summary measures, synthesis of results (e.g. meta-analysis, best evidence synthesis) and risk of bias across studies are not described. Looking at the results, meta-analysis to calculate SMD has been performed, and there is mention of limited and moderate evidence. But these have not been defined or described in the method.

We modified the Methods section following the PRISMA methodology checklist, pages 3, 4, and 5. We also included the information of the heterogeneity analysis in Data synthesis, page 5.

8. In line with comment 7 above, meta-analysis was undertaken but there does not appear to be an assessment of the risk of bias across studies or overall confidence of the evidence (e.g. the GRADE approach). This should be done for each meta-analysis comparison to provide insight on the quality of meta-analysis evidence and confidence in the findings.

We assessed the risk of bias using the Cochrane Risk of Bias Tool, as described on page 5 and S3 Table. 

Results:

9. Paragraph 1 – Science direct and research gate are mentioned here, but were not mentioned in the Method. Were these used as additional databases for the search?

We obtained the articles from Science Direct and Research Gate from the references of some studies included in the previous review. However, based on the suggestion of expanding our sources, we have included the Science Direct database in this second review.

10. Reporting of the Results would be much clearer if it were arranged using the sections defined in the Prisma checklist. For example, the section labelled Evidence Hierarchy appears to detail the Study Characteristics. Risk of Bias of individual studies is currently reported in the Method section, so should be presented in the Results instead. Splitting the reporting of results into sections detailing results of individual studies, then detailing the synthesis of results (i.e. meta-analysis and qualitative synthesis) in accordance with the Prisma guidelines would be a much neater way for arranging the Results.

Thanks for the feedback. As suggested, we report our results using the sections of the PRISMA checklist. We present the findings for each type of comparison and then the meta-analysis of each one. Pages 5 to 24.

11. The paper is largely well-written, however there are a few grammatical errors (e.g. LLLT is defined twice on page 6), so I suggest reviewing the manuscript for such issues.

We followed the grammatical suggestions and named the modalities with acronyms only at the beginning.

12. Forest plots – suggest adding labels to the X axis to show which direction favours the treatment and which direction favours controls

We add the label on the x-axis of each forest plot indicating the direction of favourability of the treatments. 

13. Meta-analysis text results – suggest including I-squared when reporting the SMD meta-analysis results, to show heterogeneity without needing to refer to the supplementary material.

We included in each forest plot the heterogeneity results of the I2 tests. 

Discussion:

14. The discussion is largely a summary of the results. It would be strengthen the paper to discuss how these findings relate to other studies, consider clinical implications etc.

We wrote the discussion in terms of clinical relevance contrasting our findings with recent reviews on the topic. 

Reviewer #2: 

The authors provide a systematic analyses of electrophysical therapy in peripheral nerve entrapment injuries. They included all papers from the course of the last decade and conclude a lack of evidence about the efficacy of electrophysical therapy.

Prisma guidelines were used to prepare the manuscript. I think it is an interesting topic to evaluate non-surgical treatment of peripheral nerve entrapments. Unfortunately the manuscript is very exhausting to read. The authors sum up every included study with a lack of detailed discussion (for example is splinting a viable option, etc.) Preclinical data about the efficacy is not discussed and compared to clinical experience.

There is a lot of information included in an enumerated fashion.

I would suggest to rewrite the manuscript to catch the reader’s attention more and meet the high standard of the journal

We appreciate your valuable feedback, which allowed us to readjust our review. We describe the results according to the comparison types, clearly stating the findings obtained for all the parameters evaluated and their respective meta-analyses. 

We rewrote the discussion contrasting our findings with those obtained in the latest reviews of the topic. We compare all the results with the references of the minimal clinically important differences of each of the evaluated parameters. We rewrote a large part of the manuscript considering the suggestions to catch the reader’s attention and meet the high standard of the journal. We hope that we have succeeded.

---

## [Decision Letter · Decision Letter 1]

30 Dec 2020

PONE-D-20-21315R1

Effectiveness of electrophysical modalities in the sensorimotor rehabilitation of radial, ulnar, and median neuropathies: a meta-analysis

PLOS ONE

Dear Dr. Bula-Oyola,

Thank you for submitting your manuscript to PLOS ONE. After careful consideration, we feel that it has merit but does not fully meet PLOS ONE’s publication criteria as it currently stands. Therefore, we invite you to submit a revised version of the manuscript that addresses the points raised during the review process.

We look forward to receiving your revised manuscript.

Kind regards,

Leila Harhaus

Academic Editor

PLOS ONE

Reviewers' comments:

Reviewer's Responses to Questions

**Comments to the Author**

1. If the authors have adequately addressed your comments raised in a previous round of review and you feel that this manuscript is now acceptable for publication, you may indicate that here to bypass the “Comments to the Author” section, enter your conflict of interest statement in the “Confidential to Editor” section, and submit your "Accept" recommendation.

Reviewer #1: (No Response)

Reviewer #2: All comments have been addressed

2. Is the manuscript technically sound, and do the data support the conclusions?

Reviewer #1: Yes

Reviewer #2: Yes

3. Has the statistical analysis been performed appropriately and rigorously? 

Reviewer #1: Yes

Reviewer #2: Yes

4. Have the authors made all data underlying the findings in their manuscript fully available?

Reviewer #1: Yes

Reviewer #2: (No Response)

5. Is the manuscript presented in an intelligible fashion and written in standard English?

Reviewer #1: Yes

Reviewer #2: Yes

6. Review Comments to the Author

Reviewer #1: Thank you for the invitation to re-review this manuscript.

I wish to thank the authors for their time and efforts revising the manuscript, I agree that it is substantially improved in its revised form. Overall, this is a thorough and comprehensive piece of work that will make a valuable contribution to the field.

Please see below some comments for consideration:

1) Abstract: ‘We found limited evidence favouring therapies with extracorporeal shock-wave, pulsed radiofrequency, and low-level laser.’ It isn’t clear if this refers to limited quality of evidence, or limited effect size. Please revise to make this more apparent.

2) Abstract: missing full stop from final sentence of conclusion

3) Introduction: ‘Symptoms may include partial or total motor dysfunction of the forearm and hand, loss of muscle tone and strength, hypoesthesia or hyperesthesia, pain, allodynia, or paraesthesia’ A minor point, these are both signs and symptoms, so please start this sentence with ‘Signs and symptoms may include…’

4) Methods: For clarity in the Data Synthesis section, please state explicitly that this refers to the meta-analysis performed. Also, some comment on interpretation of effect sizes (small, large etc) would be good, and this would help with the clinical interpretation of findings, and could be integrated/mentioned through the reporting of results (or at least in the Discussion).

5) Methods: The authors have used Cochrane Risk of Bias Tool for assessment of Risk of bias in individual studies (nicely shown in Table S3), however there is no assessment made of risk of bias across studies (like the GRADE approach) to consider confidence in the findings of the meta-analysis comparisons. This would strengthen the review and the manuscript if possible to include.

6) Results: Table S4 – the authors should be commended on the time and work required to produce this comprehensive summary of the results of all included findings. A minor note, it is a bit confusing having Appendix 1 written inside Table S4, suggest just using the term Table S4 rather than Appendix 1

7) Results: Paragraph 1 in subheading ‘Effects of electrophysical interventions’ – Is there a parenthesis missing somewhere?

8) Results: Forest plots – there are instances of a paper appearing twice within the same forest plot. Why is this? It should be made clearer why this is the case in the caption of each figure.

9) Results: I like the comparison of the meta-analysis findings with MCID, that is a nice way to help convey the clinical significance of the findings. Was this done using the mean differences for the outcomes (rather than SMDs) calculated in the meta-analysis? If so, it would be helpful for these data to be included (even just reported as a range in the text) for the outcomes mentioned in the Clinical Significance paragraph (p25, paragraph 1).

10) Discussion: the Discussion succinctly compares findings of the present review and considers the clinical relevance of findings, with respect to the MCID as per comment 9. As noted in comment 4 above, discussing the effect sizes reflected by the SMDs would be another way to explore the clinical significance of these results (e.g. small effect, large effect etc).

11) Discussion: How much confidence do the authors have in the results of their meta-analysis comparisons? The discussion of the results largely indicate that electrophysical modalities do not produce clinically significant treatment effects. Considering the quality of the studies that comprise the meta-analysis (plus numbers of participants, effect sizes etc), is this ‘good quality’ meta-analysis data? The GRADE approach referred to in comment 5 is a nice way to consider this should the authors elect to utilise it.

12) Figures: there are a lot of figures, but they are well done and clear.

Reviewer #2: The authors made a substantial effort top improve their manuscript. The data presentation and discussion gained readability.

I found some minor potential for improvement:

Line 165n and ongoing paragraph should be written in complete sentences and not only in a abbreviated manner.

Figure 9: All other figures show the sham procedure on the right side of the figure this one on teh left. This might be confusing for the Reader

Figure 12: same

7. PLOS authors have the option to publish the peer review history of their article (what does this mean?). If published, this will include your full peer review and any attached files.

Reviewer #1: No

Reviewer #2: No

---

## [Author Response · Author response to Decision Letter 1]

13 Feb 2021

February 13, 2021

Reviewers

Plos One

Dear editor and reviewers:

Thank you for this new opportunity to review our manuscript. We appreciate your thoughtful comments which have contributed to the substantial improvement of our revision. We present our responses in bold after each comment and highlight changes to the manuscript in blue. We believe that we have improved the methods of evidence assessment and the discussion of results. We hope that this revised version reflects all comments and is better suited to your journal for publication.

Reviewer #1: Thank you for the invitation to re-review this manuscript.

I wish to thank the authors for their time and efforts revising the manuscript, I agree that it is substantially improved in its revised form. Overall, this is a thorough and comprehensive piece of work that will make a valuable contribution to the field.

We are grateful for the reviewer's insightful comments. We made every effort to address the reviewer's suggestions throughout the manuscript.

Please see below some comments for consideration:

1) Abstract: ‘We found limited evidence favouring therapies with extracorporeal shock-wave, pulsed radiofrequency, and low-level laser.’ It isn’t clear if this refers to limited quality of evidence, or limited effect size. Please revise to make this more apparent.

As suggested, we describe our results based on effect sizes and MCID.

2) Abstract: missing full stop from final sentence of conclusion

We have included the missing period at the end of the sentence.

3) Introduction: ‘Symptoms may include partial or total motor dysfunction of the forearm and hand, loss of muscle tone and strength, hypoesthesia or hyperesthesia, pain, allodynia, or paraesthesia’ A minor point, these are both signs and symptoms, so please start this sentence with ‘Signs and symptoms may include…’

We start the paragraph with the phrase ‘Signs and symptoms may include…’

4) Methods: For clarity in the Data Synthesis section, please state explicitly that this refers to the meta-analysis performed. Also, some comment on interpretation of effect sizes (small, large etc) would be good, and this would help with the clinical interpretation of findings, and could be integrated/mentioned through the reporting of results (or at least in the Discussion).

We rewrite the data synthesis section detailing the methods for estimating the effect sizes of the meta-analysis results. This information was also included in the results and discussion.

5) Methods: The authors have used Cochrane Risk of Bias Tool for assessment of Risk of bias in individual studies (nicely shown in Table S3), however there is no assessment made of risk of bias across studies (like the GRADE approach) to consider confidence in the findings of the meta-analysis comparisons. This would strengthen the review and the manuscript if possible to include.

We assess the quality of the evidence through the GRADE approach. This information was included in the results, the discussion and the summary of the findings are present as supplementary material in Table S3.

6) Results: Table S4 – the authors should be commended on the time and work required to produce this comprehensive summary of the results of all included findings. A minor note, it is a bit confusing having Appendix 1 written inside Table S4, suggest just using the term Table S4 rather than Appendix 1

We replace the term appendix by the name of the table. (Now is named as Table S5).

7) Results: Paragraph 1 in subheading ‘Effects of electrophysical interventions’ – Is there a parenthesis missing somewhere?

We have included the missing parentheses. Thank you for your willingness to improve the details.

8) Results: Forest plots – there are instances of a paper appearing twice within the same forest plot. Why is this? It should be made clearer why this is the case in the caption of each figure.

We have included the clarifying information as a caption on the image. We list studies that compared more than one treatment modality or dose.

9) Results: I like the comparison of the meta-analysis findings with MCID, that is a nice way to help convey the clinical significance of the findings. Was this done using the mean differences for the outcomes (rather than SMDs) calculated in the meta-analysis? If so, it would be helpful for these data to be included (even just reported as a range in the text) for the outcomes mentioned in the Clinical Significance paragraph (p25, paragraph 1).

We describe the information about the MCID estimate and include a reference table with the values.

10) Discussion: the Discussion succinctly compares findings of the present review and considers the clinical relevance of findings, with respect to the MCID as per comment 9. As noted in comment 4 above, discussing the effect sizes reflected by the SMDs would be another way to explore the clinical significance of these results (e.g. small effect, large effect etc).

We reinforced the discussion about the clinical significance of the results by contrasting them with effect sizes and MCID

11) Discussion: How much confidence do the authors have in the results of their meta-analysis comparisons? The discussion of the results largely indicate that electrophysical modalities do not produce clinically significant treatment effects. Considering the quality of the studies that comprise the meta-analysis (plus numbers of participants, effect sizes etc), is this ‘good quality’ meta-analysis data? The GRADE approach referred to in comment 5 is a nice way to consider this should the authors elect to utilise it.

We include the assessment of the quality of the evidence as an opening for the discussion and highlighting its implications in the effect estimation.

12) Figures: there are a lot of figures, but they are well done and clear.

We appreciate your feedback.

Reviewer #2: The authors made a substantial effort to improve their manuscript. The data presentation and discussion gained readability.

Thanks for your valuable comments.

I found some minor potential for improvement:

Line 165n and ongoing paragraph should be written in complete sentences and not only in a abbreviated manner.

We replace abbreviations with full names of treatments and neuropathies.

Figure 9: All other figures show the sham procedure on the right side of the figure this one on the left. This might be confusing for the Reader. Figure 12: same

As suggested, we modify the forest plots (Figure 9 and 12).

Sincerely, 

Ena Lucía Bula Oyola

Universidad del Norte

Universitat Politècnica de València 

+34 667045622

oyolae@uninorte.edu.co

---

## [Decision Letter · Decision Letter 2]

1 Mar 2021

Effectiveness of electrophysical modalities in the sensorimotor rehabilitation of radial, ulnar, and median neuropathies: a meta-analysis

PONE-D-20-21315R2

Dear Dr. Bula-Oyola,

We’re pleased to inform you that your manuscript has been judged scientifically suitable for publication and will be formally accepted for publication once it meets all outstanding technical requirements.

Kind regards,

Leila Harhaus

Academic Editor

PLOS ONE

Reviewers' comments:

Reviewer's Responses to Questions

**Comments to the Author**

1. If the authors have adequately addressed your comments raised in a previous round of review and you feel that this manuscript is now acceptable for publication, you may indicate that here to bypass the “Comments to the Author” section, enter your conflict of interest statement in the “Confidential to Editor” section, and submit your "Accept" recommendation.

Reviewer #1: All comments have been addressed

Reviewer #2: All comments have been addressed

2. Is the manuscript technically sound, and do the data support the conclusions?

Reviewer #1: Yes

Reviewer #2: Yes

3. Has the statistical analysis been performed appropriately and rigorously? 

Reviewer #1: Yes

Reviewer #2: Yes

4. Have the authors made all data underlying the findings in their manuscript fully available?

Reviewer #1: Yes

Reviewer #2: Yes

5. Is the manuscript presented in an intelligible fashion and written in standard English?

Reviewer #1: Yes

Reviewer #2: Yes

6. Review Comments to the Author

Reviewer #1: Thank you for the invitation to re-review this paper. The authors have done a great job addressing the second round of review comments. I agree the paper has been substantially improved.

I have no further concerns regarding the manuscript and would like to commend the authors on their comprehensive study. I note however that there are currently two different files named Table S3 (one is a replication of Table S4?), however I imagine this will be sorted out at a later stage in the editing process.

Reviewer #2: I think the authors again made an effort to improve their manuscript which is now publishable from my opinion

7. PLOS authors have the option to publish the peer review history of their article (what does this mean?). If published, this will include your full peer review and any attached files.

Reviewer #1: No

Reviewer #2: No

---

## [Editor Report · Acceptance letter]

5 Mar 2021

PONE-D-20-21315R2 

Effectiveness of electrophysical modalities in the sensorimotor rehabilitation of radial, ulnar, and median neuropathies: a meta-analysis 

Dear Dr. Bula-Oyola:

I'm pleased to inform you that your manuscript has been deemed suitable for publication in PLOS ONE. Congratulations! Your manuscript is now with our production department. 

Kind regards, 

on behalf of

Prof. Dr. med. Leila Harhaus 

Academic Editor

PLOS ONE